# Prominent Josephson tunneling between twisted single copper oxide planes of $Bi_2Sr_{2-x}La_xCuO_{6+y}$

Heng Wang [1], Yuying Zhu [2,3] ✉, Zhonghua Bai[1], Zechao Wang [4,5], Shuxu Hu[1], Hong-Yi Xie[2], Xiaopeng Hu[1], Jian Cui[2], Miaoling Huang[2], Jianhao Chen [2,6], Ying Ding[7], Lin Zhao [7], Xinyan Li [7], Qinghua Zhang[7], Lin Gu[4,7], X. J. Zhou[7], Jing Zhu [4,5], Ding Zhang [1,2,8] ✉ & Qi-Kun Xue[1,2,9] ✉

Josephson tunneling in twisted cuprate junctions provides a litmus test for the pairing symmetry, which is fundamental for understanding the microscopic mechanism of high temperature superconductivity. This issue is rekindled by experimental advances in van der Waals stacking and the proposal of an emergent $d+id$-wave. So far, all experiments have been carried out on $Bi_2Sr_2CaCu_2O_{8+x}$ (Bi-2212) with double $CuO_2$ planes but show controversial results. Here, we investigate junctions made of $Bi_2Sr_{2-x}La_xCuO_{6+y}$ (Bi-2201) with single $CuO_2$ planes. Our on-site cold stacking technique ensures uncompromised crystalline quality and stoichiometry at the interface. Junctions with carefully calibrated twist angles around 45° show strong Josephson tunneling and conventional temperature dependence. Furthermore, we observe standard Fraunhofer diffraction patterns and integer Fiske steps in a junction with a twist angle of 45.0±0.2°. Together, these results pose strong constraints on the $d$ or $d+id$-wave pairing and suggest an indispensable isotropic pairing component.

The recent application of van der Waals (vdW) stacking technique to high temperature ($T_c$) cuprate superconductors[1–5] revives a dormant debate over the pairing symmetry of the order parameter[6–8]. Phase sensitive experiments using in-plane Josephson junctions showed evidence for $d$-wave pairing nearly three decades ago[9,10]. However, experiments[11–13] of the $c$-axis Josephson effect, which is also able to determine the phase component of the pairing, revealed a large supercurrent between two cuprate crystals at a twist angle of 45°. The results were against the $d$-wave pairing scenario[14,15] and suggested isotropic pairing instead. Since Josephson tunneling depends sensitively on the details of the interface, poor sample quality may account for the inconsistency[16]. Of late, the vdW stacking of two-dimensional materials allows ever-expanding material pool[17,18] with improved interface quality[19–21] and precision control[22,23]. These advancements benefit the study of high-$T_c$ superconductivity too[1–4,24,25], because cuprates such as Bi-2212 are layered materials that can be exfoliated, even down to the monolayer level[1,5]. Twisted Bi-2212 Josephson junctions are successfully realized by restacking the ultrathin flakes[6–8]. They show atomically flat interfaces, closely matching the theoretically considered situation. This advancement

[1]State Key Laboratory of Low Dimensional Quantum Physics and Department of Physics, Tsinghua University, Beijing 100084, China. [2]Beijing Academy of Quantum Information Sciences, Beijing 100193, China. [3]Hefei National Laboratory, Hefei 230088, China. [4]National Center for Electron Microscopy in Beijing, School of Materials Science and Engineering, Key Laboratory of Advanced Materials (MOE), The State Key Laboratory of New Ceramics and Fine Processing, Tsinghua University, Beijing 100084, China. [5]Ji Hua Laboratory, Foshan, Guangdong 528200, China. [6]International Center for Quantum Materials, School of Physics, Peking University, Beijing 100091, China. [7]Institute of Physics, Chinese Academy of Sciences, Beijing 100190, China. [8]RIKEN Center for Emergent Matter Science (CEMS), Wako, Saitama 351-0198, Japan. [9]Southern University of Science and Technology, Shenzhen 518055, China. ✉e-mail: zhuyy@baqis.ac.cn; dingzhang@mail.tsinghua.edu.cn; qkxue@mail.tsinghua.edu.cn

allows a revisit to the observation of isotropic pairing in twisted cuprates[6].

The experimental progresses have also instigated fresh theoretical efforts[26–32]. At the twist angle of 45°, while the complete suppression of Josephson tunneling is demanded by pure $d$-wave symmetry, it is recently proposed that a $d+id$ or $d+is$ order parameter[27] can account for the finite Josephson tunneling. This emergent order parameter can open up a nodeless gap[27] and gives rise to a non-monotonic temperature dependence of the Josephson critical current $I_c(T)$ or critical current density $J_c(T)$[28,29]. The Fraunhofer diffraction pattern is expected to show doubling in frequency or halving of the period, owning to the co-tunneling of paired Cooper pairs with a net charge of $4e$ instead of $2e$[26,28,29]. Theory also predicts that the 45°-twist bilayer can produce Shapiro steps at half-integer positions under microwave irradiation. Although the proposed full gap was not observed in post-annealed junctions[6], an experiment using cryogenically fabricated junctions exhibited non-monotonic $I_c(T)$ and half-integer Shapiro steps[7], hinting at exotic pairing. Moreover, there is an apparent inconsistency in the angular dependences of the Josephson coupling strength $I_cR_n$ ($R_n$ is the normal state resistance) in these experiments. Further experimental and theoretical investigations[30,31] are therefore necessary to elucidate these discrepancies.

So far, all the existing experiments on $c$-axis twisted cuprates have been carried out on a single type of cuprate superconductors—Bi-2212[6–8,11–13]. Whether the anomalous tunneling phenomena can be extended to other high temperature cuprate superconductors remains unaddressed. In this work, we choose Bi-2201 to fabricate twisted cuprates and study the Josephson tunneling (Fig. 1a). Although a series of scanning tunneling and photoemission experiments have been carried out on Bi-2201[33–35], this material has not been employed in either the in-plane or out-of-plane Josephson tunneling experiments for phase-sensitive detection. Bi-2201 has the advantage that it avoids the unnecessary complexity brought by the bonding and antibonding effects of double $CuO_2$ layers in Bi-2212[36,37]. While reduction from

bilayer to single $CuO_2$ layer enhances the two-dimensional effect[33], this situation better matches that considered by the theories[27]. Technically, the doping level of ultrathin Bi-2201 can be more stable since it is governed by La substitution of Sr and interstitial O. The stoichiometry can be better preserved during the sample fabrication process[1,3,5]. In the case of Bi-2212, the junctions often host a relatively short Josephson penetration depth $\lambda_j$ ($\sim$0.2 μm) in comparison to the typical junction width (>1 μm). It gives rise to nonuniform phase distribution[38,39], which may further induce fractional Shapiro steps[40]. To enhance uniformity, one can use the underdoped Bi-2212 with reduced $J_c$ (<100 A/cm$^2$) such that $\lambda_j$ >1 μm[39,41]. Since Bi-2201 hosts a similarly reduced $J_c$, it can be employed as the alternative route[42–45] (we provide further estimation in Supplementary Note 1).

Here, we fabricate multiple twisted Bi-2201 junctions with their twist angles calibrated to be close to 45°. We demonstrate their atomically flat interfaces and uncompromised stoichiometry by atomically resolved structural analysis and low temperature transport studies. All these junctions show prominent Josephson tunneling, indicating a systematic deviation from the expected behavior of $d$-wave pairing. We further reveal the conventional $I_c(T)$ and the standard Fraunhofer pattern in one of the junctions with a calibrated twist angle of $45.0 \pm 0.2°$. The absence of frequency-doubling as a function of temperature, together with the integer Fiske steps, strongly speaks against the existence of a subdominant Cooper pair co-tunneling component—a crucial ingredient of $d+id$-wave pairing. Our study establishes that there exists pronounced isotropic pairing component in the 45°-twist cuprate bilayers.

## Results
### Device fabrication and structural analysis
We employ a carefully designed on-site cold stacking method to fabricate Bi-2201 junctions[3,5–7]. The whole fabrication process (Fig. 1b) assures that the cleaved flakes never leave the cold stage before protection-layer capping. As such, the crystallinity and

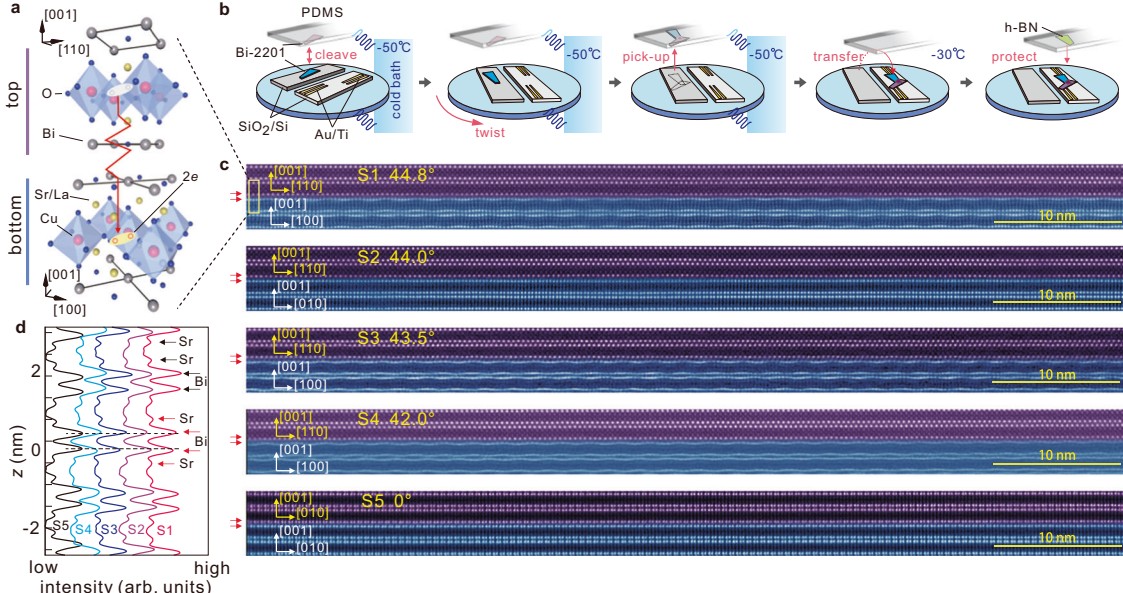

**Fig. 1 | Twisted Josephson junctions of Bi-2201. a** Schematic structure of the junction at twist angle of 45°. The section marked as top (bottom) consists of half a unit cell in the $c$-axis. **b** Key steps of the on-site cold stacking: cleaving, twisting, up-picking, releasing and h-BN protection. The first panel illustrates the constituent components. The PDMS and the sample holder (round disk) are thermally anchored to the cold bath throughout the fabrication steps. **c** Cross-sectional high-angle annular dark field scanning transmission electron microscopy (HAADF-STEM) image of samples S1 to S5 around the twisted interface region. Purple and blue colors highlight the top and bottom Bi-2201, respectively. The twisted angles are indicated in the figures. Arrows indicate the BiO planes at the twisted interface. **d** Integrated intensity of the HAADF-STEM image as a function of $z$. The boundary between the top and bottom Bi-2201 is fixed at $z = 0$. Each line profile is obtained by normalizing the integrated intensity over a horizontal span of 12–21 nm. Arrows mark the peaks from BiO or SrO planes. Dashed lines indicate the positions of BiO planes at the interface for sample S5.

stoichiometry of the bulk are well preserved in the artificial junctions. Figure 1b shows that the substrate holder and the cantilever with the polydimethylsiloxane (PDMS) stamp are cooled to a temperature below −50 °C. Two SiO$_2$/Si substrates−one bare and another with prepatterned electrodes−are anchored on the same holder. We first cleave a Bi-2201 flake with a typical thickness of 300 nm on the bare substrate into two pieces. The flake left on the substrate, which is usually thicker, is subsequently rotated against its thinner counter-part on the stamp. We then use the stamp to pick up the rotated flake by making sure that it has a small overlap with the initially cleaved flake. The overlapping region therefore constitutes the twisted arti-ficial junction. An ultraclean interface is guaranteed because there exists no glue residue from the tape or the stamp on the two freshly cleaved surfaces. The whole stack consisting of both top and bottom Bi-2201 is consequently transferred onto the other substrate with prepatterned electrodes. The release procedure is taken at about −30 °C. Finally, a flake of hexagonal boron nitride (h-BN) was deposited onto the junction region for protection.

Figure 1c demonstrates the high quality of our junctions. We obtain atomically resolved structures (after the transport measure-ments) in a large spatial span of about 70 nm. Images with equiva-lently high quality were taken from multiple locations along the direction parallel to the interface. The artificial interface, consisting of the lowest BiO layer of the top Bi-2201 and the uppermost BiO layer of the bottom Bi-2201 (indicated by arrows), patently distin-guishes itself from other double BiO layers in either side of Bi-2201. For samples S1, S3 and S4, wavy undulations can be observed in the bottom halves of the images, suggesting that these sections are viewed from the [0$\bar{1}$0] crystalline direction perpendicular to the uniaxial supermodulations running along the [100] direction. The top parts show no wavy structures because they are viewed from the [1$\bar{1}$0] direction that is along the diagonal between [0$\bar{1}$0] and [100]. The supermodulations are usually attributed to the misfit between the CuO$_2$/SrO planes and the BiO plane[46,47] and can locally modulate the energy gap[48]. At the twisted interface, only one of the two BiO layers shows the persisting supermodulation. For sample S2, we observe that the Bi atoms in the bottom Bi-2201 bunch into pairs whereas the top Bi-2201 shows no such behavior. It indicates that the bottom part is viewed from [100], which is orthogonal to that of the bottom halves of S1, S3, and S4. At the interface, only one of the two BiO layers shows bunching. For sample S5, both the top and bottom Bi-2201 show bunching but the bunching in the top Bi-2201 is shifted by half a period in comparison to the bottom Bi-2201. Notably, the twist angles of samples S1 and S2 must be very close to 45°. This can be immediately inferred from the simultaneous atomic resolution of both the top and bottom parts of Bi-2201 with distinct crystalline orientations. A more precise determination by using the Kikuchi patterns (details in Supplementary Note 2 and Supplementary Figs. 1–2)[6] yields a deviation ($\Delta\theta$) from 45° as small as 0.2° for S1 and 1.0°, 1.5°, 3.0° for S2, S3, and S4. For simplicity, we use $\theta = 45° − \Delta\theta$ as the twist angles for these samples throughout the paper, since $\theta = 45° + \Delta\theta$ is equivalent.

Figure 1d shows the normalized intensity along the c-axis for samples S1 to S5. One sees that the BiO and SrO planes (red arrows) at the junction interface possess comparable integrated intensities to those of the bulk (black arrows). More importantly, the two rotated CuO$_2$ planes exhibit the overlapping signal intensities of Cu atoms as those CuO$_2$ planes far away from the interface (Supplementary Fig. 3), attesting to the uncompromised quality of the two superconducting layers involved in the Josephson tunneling process. We note that the distance between the two nearest-neighboring BiO layers ($d_{Bi−Bi}$) at around $z = 0$ is larger in samples S1 to S4 than that of sample S5. The corresponding distance between the two CuO$_2$ planes ($d_{Cu−Cu}$) across the interface also increases by about 5% from the value in S5 to that in S1−S4 (Supplementary Fig. 4).

## Junction resistance and tunneling

We now study the transport properties in the same batch of samples (S1−S4) with their crystalline structures well clarified. All junctions exhibit a single superconducting transition (Fig. 2a1−a4 and Supple-mentary Fig. 5). For S3, we further demonstrate that the transition temperature of the junction is exactly the same as that measured from the individual flakes. These results demonstrate that the doping con-centration is preserved throughout the complete stack of Bi-2201 flakes. Figure 2b1−b4 presents the tunneling I-V characteristics. These samples possess a single branch of supercurrent before transitioning to the normal state. In Supplementary Figs. 6−7, we show the I-V characteristics for two samples in a wide current range. The single jump in one current direction clearly distinguishes from those hys-teretic I-V behaviors caused by phase slip lines[49,50]. We argue that the observed Josephson tunneling solely occurs between the two twisted CuO$_2$ planes since they have the smallest tunneling area (numbers listed in Fig. 2b1−b4). The intrinsic junctions in the bottom Bi-2201, connected in series in the electrical circuit, have much larger tunneling area (3–10 times larger) such that their critical current is not reached in our measured regime (further support is given in Supplementary Notes 3 and 4 and Supplementary Fig. 7). For samples S3, S4 and OP1, the I-V characteristics show jumps from the zero-bias branch to the branch with finite resistances. There also exists prominent hysteresis in the two opposite sweeping directions. For samples S1 and S2 (Sup-plementary Fig. 5b), however, the switching seems continuous. These two samples are patterned by focused ion beam (FIB) into long strips (insets of Fig. 2a and Supplementary Fig. 5a) for the investigation of Fraunhofer patterns (to be discussed in the following section). This additional step seems to suppress the hysteresis. Nevertheless, we will show that data from these two samples also stand against the expec-tation from either d- or d+id-wave pairing. Figure 2c1−c4 shows that our samples exhibit the conventional temperature dependence of $I_c$: a rapid rise of $I_c$ when T just decreases from $T_c$ followed by saturation of $I_c$ at low temperatures ($T \leq 0.5 T_c$). This is similar to that prescribed by the Ambegaokar-Baratoff (AB) formula for the Josephson tunneling between two conventional superconductors[51]. By contrast, the recent theories for d+id -wave pairing[28,29] predict a gradual increase of $I_c$ from $T = T_c$ to about $0.5 T_c$ followed by a pronounced rise at lower T at twist angles close to 45°. They also predict strong non-monotonic $I_c(T)$ around 30°. These features distinguish the higher-order term due to co-tunneling of Cooper pairs from the standard Josephson term. The conventional $I_c(T)$ observed in samples S1-S4 and OP1 therefore sug-gest the dominance of the first-order Josephson tunneling process.

## Fraunhofer pattern and Fiske steps

Further insight into the Josephson effect in twisted bilayer cuprates can be garnered from the magnetic field dependence. For sample S1, we measure the Josephson critical current as a function of magnetic field $B_\parallel$ applied parallel to the longer direction of the junction (Fig. 3a). Figure 3b displays the critical Josephson currents at zero bias from the $I − V$ characteristics at different $B_\parallel$, showing the well-defined Fraun-hofer diffraction pattern. Figure 3c further shows the colored plot of $dI/dV$ as a function of current and $B_\parallel$ at different temperatures. It captures both the Fraunhofer pattern as well as the AC Josephson effect. Solid curves in the figures are theoretical fits by using the standard formula: $I_c = I_{c,0} |\sin\Phi/\Phi|$, where $I_{c,0}$ is the zero-field critical current and $\Phi = \pi B_\parallel/B_0$ with $B_0$ being the oscillation period. This standard formula nicely reproduces the experimental features. By contrast, if the d+id-wave is present as a subdominant term, the co-tunneling of Cooper pairs (with 4e) will cause suppression of $I_c(B_\parallel)$ at $nB_0/2$, with $n = 1, 3, 5\cdots$ (dashed curves in Fig. 3)[26–29]. This frequency-doubling effect is expected to be most pronounced at low tempera-tures and gets smeared out when $T \rightarrow T_c$. Such a phase transition is clearly absent in S1 here and in S2 in Supplementary Fig. 5. Instead, we observe only a subtle change, as indicated by the decreasing value of

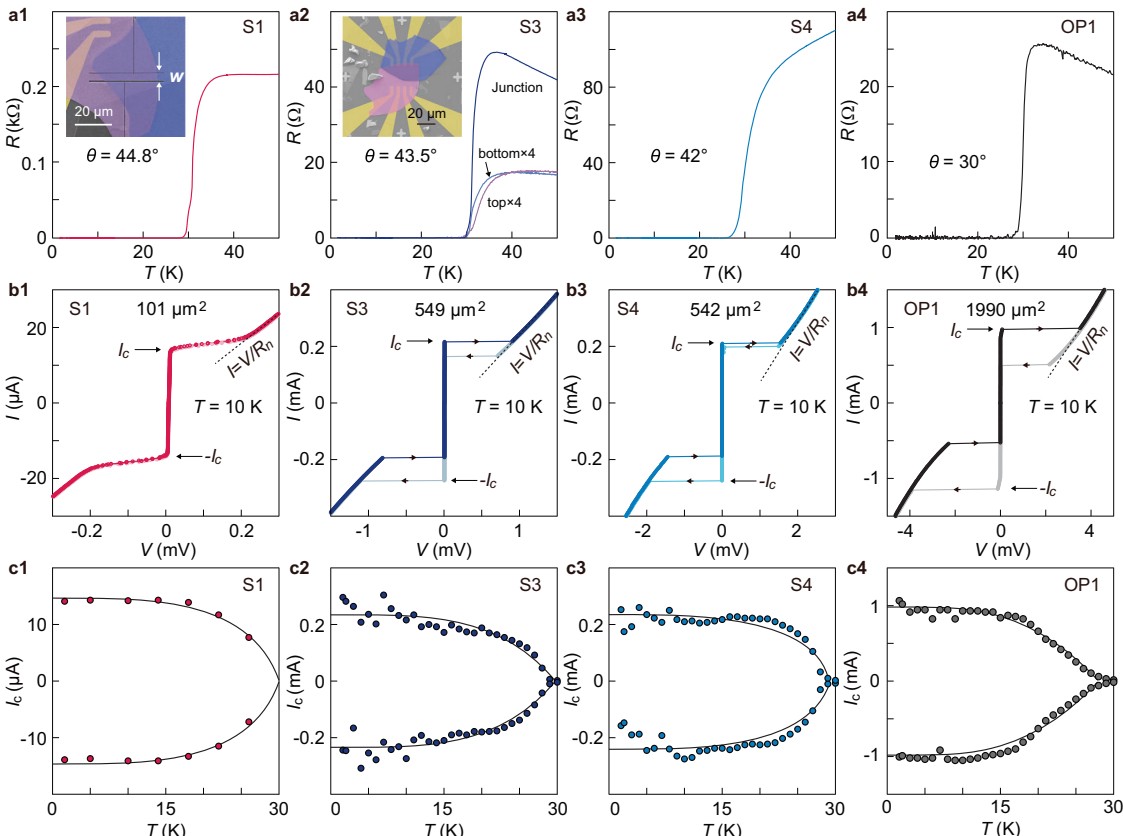

**Fig. 2 | Transport characterizations of the twisted junctions. a1–a4** Temperature dependent junction resistance, indicating a single superconducting transition. Insets in (**a1**) and (**a2**) are false-colored SEM images of the respective samples. Sample S1 is patterned by focused ion beam (FIB). The arrows mark the width of junction ($w = 3.5$ μm). For S3, resistances of individual flakes are also provided (marked as top/bottom). **b1–b4**, $I − V$ characteristic across the junction at 10 K. Numbers listed in the middle indicate junction areas. Short arrows indicate the sweeping directions. Long arrows mark the Josephson critical currents. Dotted lines show the slopes of the normal state, which are used for determining the normal state resistance. **c1–c4** Temperature dependence of the Josephson critical current. Black curves are guide to the eye. We use a modified Ambegaokar-Baratoff formula[6] to generate the curves.

$B_0$ with increasing $T$. We defer further discussions on this evolution to the end of the paper. In addition, noises in the color plot become suddenly prominent upon increasing $B_\parallel$ to a certain value (marked by the red arrows). This onset field becomes larger with decreasing temperatures, reminiscent of the temperature dependence of critical magnetic field (Supplementary Fig. 8). We therefore attribute the noises to the entrance of Abrikosov vortices above the lower critical field. They are created by a small out-of-plane magnetic field due to slight misalignment between the magnetic field and the sample plane.

Notably, there exist additional regular branches in the Fraunhofer patterns (marked by horizontal arrows). Between these branches and the major lobe of the Fraunhofer pattern (following the solid curves) are regions hosting local minima of $dV/dI$ (dark blue color). These regions are in fact related to the AC Josephson effect. Typically, the AC Josephson effect is demonstrated by current steps at discrete voltages −Shapiro steps−in a junction under microwave irradiation. Here, no external microwave source is necessary because we utilize the rectangular shape of our junction. Such a cavity possesses certain electromagnetic modes which can be excited under a bias voltage. Interference between the electromagnetic fields resonating in the cavity and the AC Josephson effect of the junction gives rise to another type of current steps at discrete voltages−Fiske steps (inset of Fig. 4a)[52]. Conventionally, Fiske steps occur at $V_k = k\Phi_0 f_c$, where $k = 1,2,3\cdots$, $\Phi_0 = h/2e$ is the flux quantum and $f_c$ is the fundamental frequency of the junction cavity mode. We replot the data in Fig. 3 in terms of d$I$/d$V$ as a function of the junction voltage and $B_\parallel$ in Fig. 4b and c. A Fiske step in $V(I)$ (Fig. 4a) gives rise to a local peak in d$I$/d$V$ (Fig. 4b). Ten

horizontal stripes at non-zero voltages, reflecting ten Fiske steps, can be identified in Fig. 4c (marked alphabetically: $a, b, c, d, e$). The Fiske steps observed in sample S2 are shown in Supplementary Fig. 9. For each Fiske step, its intensity gets modulated by the magnetic field and breaks into several sections, as indicated by the black, gray and white triangles. The maximal intensity in each section drops monotonically with increasing $B_\parallel$. The dotted curve in Fig. 4d reflects this monotonic behavior of the first step (stripe-$a$). Notably, the maximal intensity of the first section from each stripe also shows a monotonic decreasing trend with the voltage position (dashed curve in Fig. 4d). These features are all consistent with the observation in intrinsic junctions of Bi-2212[52]. It indicates that the Fiske steps follow a continuously increasing integer of $k$ from stripe-$a$ to stripe-$e$. Interestingly, the departure from stripe-$a$ to zero bias ($V_{a0}$) is two times the separation between the stripes ($V_{ba}$ for example). We therefore assign the stripes ($a, b, c, d, e$) as Fiske steps with $k = 2, 3, 4, 5, 6$. In this case, the first step ($k=1$) is missing at small $B_\parallel$. However, it is worth noting that the first step seems to reemerge at higher fields. This is illustrated in Fig. 4e where peaks in d$I$/d$V$ occur at the expected positions of $V = \pm V_{a0}/2$. This reentrant behavior was recently reported in Al/InAs junctions too, but in the topologically trivial regime[53]. The missing of the first step and its reemergence may be caused by the smearing effect of the resistive branch in the $I$-$V$ characteristics[54]. The strong non-linearity of the $I$-$V$ curve causes the highest dissipation at bias voltages close to zero. This can be seen from the darker color close to zero bias in Fig. 4b and c. With increasing $B_\parallel$, the normal state branch becomes less resistive such that the first step can re-appear. We point out that the monotonic increase

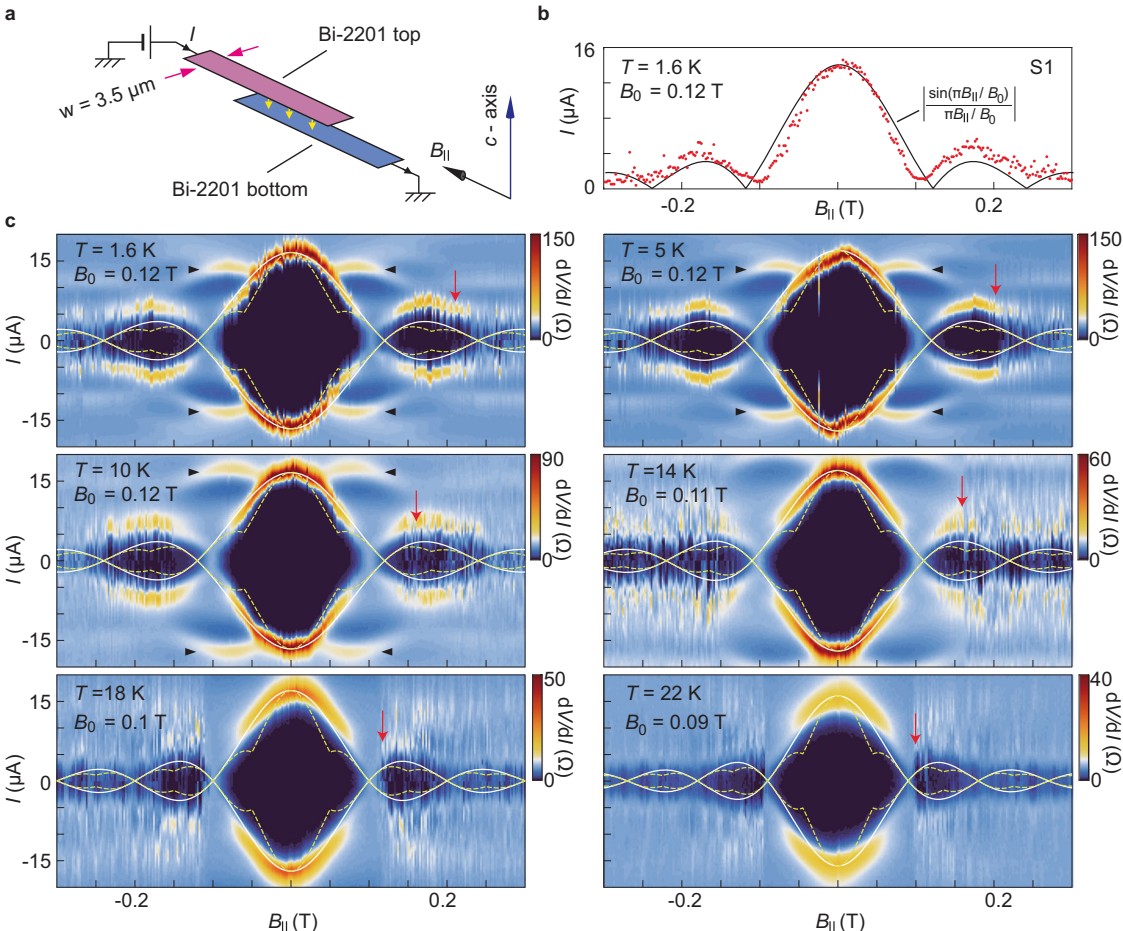

**Fig. 3 | Fraunhofer patterns of the twisted Bi-2201 junction. a** Schematic illustration of the configuration for measuring the Fraunhofer pattern. **b** Critical Josephson current as a function of in-plane magnetic field ($B_\parallel$) for sample S1. Solid curve is a theoretical fit. **c** Color-coded $dV/dI$ of sample S1 as a function of tunneling current and $B_\parallel$ at six temperature points. Red arrows mark the onset of noises. Horizontal arrows indicate the additional branch related to AC Josephson effect. Solid curves are from the formula of standard Fraunhofer pattern. $B_0$ values are the periods in the magnetic field used in plotting the solid curves. Dashed curves illustrate the doubling of oscillation frequency in the Fraunhofer pattern, as expected for $d+id$-wave pairing. Here we consider 50% contribution from the co-tunneling of Cooper pairs.

of the sequence of Fiske steps alone is compatible with Josephson tunneling of either purely $2e$ or purely $4e$ carriers. However, a dominant higher-order tunneling with $4e$ has a specific temperature dependence, which we have excluded by the conventional $I_c(T)$ in Fig. 2. The remaining possibility is that there exists a mixture of a dominant $2e$-tunneling with a small fraction of $4e$-tunneling. In this case, the Fiske steps have two sets: one prominent set at integer steps due to $2e$-tunneling and one weaker set at half integers due to $4e$-tunneling. The intensity in $dI/dV$ of the Fiske steps would therefore show non-monotonic evolution with consecutively increasing bias voltages, in sharp contrast to our data in Fig. 4.

## Discussion

In Fig. 5, we include data points from 17 samples with similar $T_c$ (within 3 K) (Supplementary Fig. 10). All the samples were fabricated by using the same recipe. We outline the upper and lower bounds of the $\cos(2\theta)$ dependence demanded by $d$-wave pairing[14,15]. They are based on the data points at 0° and 90°. Clearly, a cluster of seven data points around 45° fall outside this demarcation. The slight decrease in $I_cR_n$ at around 45°, if comparing the maximal values obtained at different angles, can be explained by the orbital effect between two conventional superconductors[55]. The observed increase of interlayer distance for S1-S4 may also play a role. Despite this variation, we highlight that these data points deviate strongly from the expectation of pure $d$-wave

pairing. Imposing a $\cos(2\theta)$ dependence, irrespective of the real angular behavior, on experimental data points sufficiently close to 45° would yield unphysically large values for $I_cR_n$ and $J_c$ at $\theta = 0°$. For example, by taking $I_cR_n(44.8°) = 0.25$ meV for S1, $I_cR_n(0°)$ has to be $0.25/\cos(2 \times 44.8°) = 35.8$ meV. This value is two times the gap of Bi-2201[56] and greatly exceeds the upper bound expected even from the AB formula. It is also one order of magnitude larger than the typical experimental $I_cR_n$ values we obtained at 0° and 90° (gray band in Fig. 5). In terms of $J_c$, a similar calculation by using the data of S2 would yield $J_c(0°) = J_c(44.0°)/\cos(2 \times 44.0°) = 1.99$ kA/cm², exceeding the measured $J_c$ in bulk Bi-2201 (Supplementary Note 4) by one order of magnitude and becoming on par with the record of Bi-2212 intrinsic Josephson junctions (Supplementary Fig. 11). We comment that $J_c$ in our junctions is actually underestimated since the effective tunneling area may be smaller than the physical area[6]. In general, a purely anisotropic $d$-wave pairing is incompatible with the experimental results, suggesting the existence of a pronounced isotropic pairing component in the twisted junctions. Such a prominent $s$-wave pairing could be a manifestation of symmetry breaking in this system. The innately present supermodulations may contribute because they break the $C4$ rotational symmetry required for the $d$-wave pairing. However, this effect was not revealed in other experiments until now.

We expand on the discussion of the temperature dependence of Josephson current $I_c(T)$. The recent theory on $d+id$-wave pairing

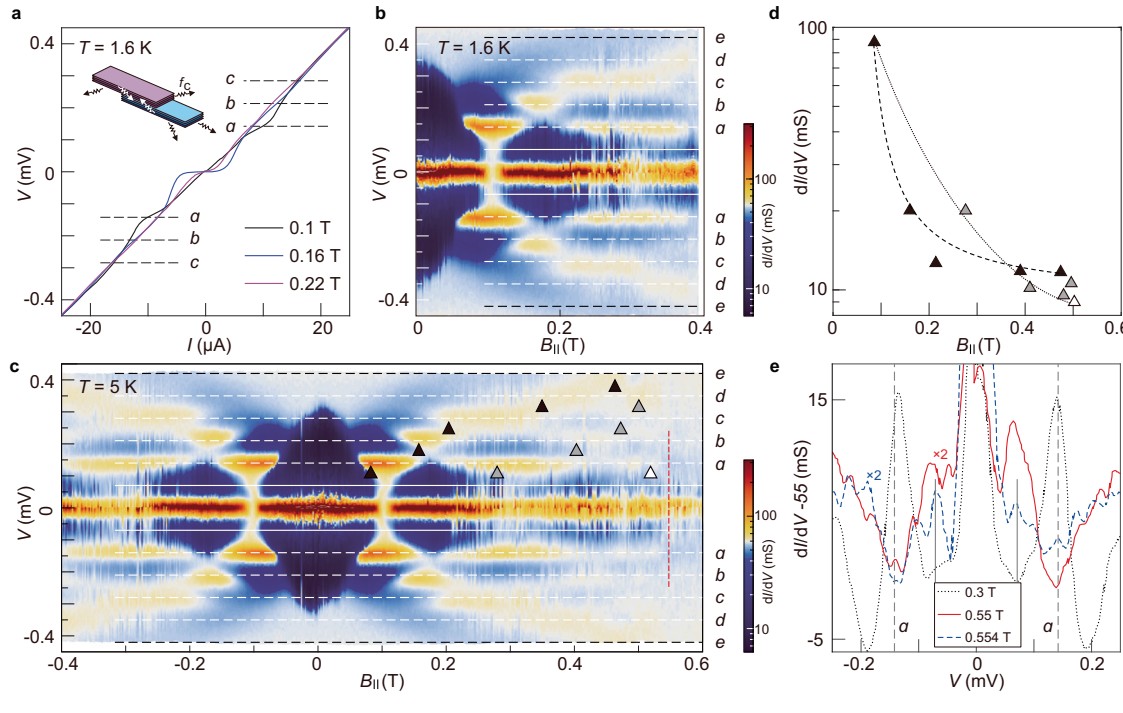

**Fig. 4 | Fiske steps in the twisted Bi-2201 junction.** Data are from sample S1. **a** Tunneling *I-V* characteristics at three in-plane magnetic fields, showing the Fiske steps at finite voltages. Dashed lines indicate the voltage positions where Fiske steps are identified in the derivative as a local peak (panel **b**). Inset is a schematic illustration of the AC Josephson effect in a cavity. The radiation from the intrinsic stacks of Bi-2201 is reabsorbed by the artificial interface, giving rise to the Fiske steps. **b**, **c** Color-coded $dI/dV$ as a function of $B_\parallel$ and the bias voltage across the junction at 1.6 and 5 K, respectively. Dashed lines mark the positions of Fiske steps (*a, b, c, d, e*). Horizontal solid lines mark the voltage positions that are half of the

values of the stripe marked as "*a*". Vertical dashed line indicates the position where the first Fiske step seems to reemerge. Triangles in (**c**) mark the positions of local maxima. **d** $dI/dV$ at the peak positions (marked as triangles in **c**) as a function of $B_\parallel$. Dashed curve indicates the continuous decreasing trend from step-*a* to step-*e*. Dotted curve indicates the decreasing trend of the same step as a function of $B_\parallel$. **e** $dI/dV$ as a function of bias voltage at three selected values of $B_\parallel$. The red and blue curves show reemergence of the first Fiske steps at the positions marked by the vertical solid lines (corresponding to the solid lines in **c**).

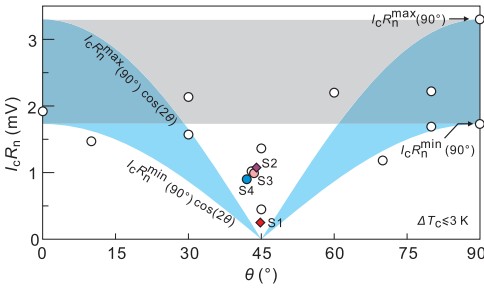

**Fig. 5 | Prominent Josephson tunneling at around 45°.** $I_c R_n$ as function of twist angle from 17 samples with their $T_c$ spreading in a narrow range of 3 K. Here samples S1 and S2 are patterned by FIB. The twist angles of S1 to S4 are determined by TEM. We use the nominal twist angles for the other data points not investigated by TEM. Blue shaded band indicates the expected behavior from pure *d*-wave scenario, if taken into account the sample-to-sample variation in $I_c R_n$ at twist angles of 90°.

proposed that the critical current hosts a Λ-shaped temperature dependence for junctions with twist angles from 18° to 36°: $I_c$ increases first as $T$ increases from zero to about $0.4 T_c$ and decreases at higher $T$. Although we obtain predominantly junctions with conventional $I_c(T)$ as shown in Fig. 2, we do observe non-monotonic $I_c(T)$ occasionally. Supplementary Figure 12a provides such an example at $\theta = 0°$. Notably, at this twist angle the theory of $d + id$-wave pairing predicts standard AB-type behavior. The angular independence suggests a more trivial origin. To shed further insight into this issue, we study the influence of

a small perpendicular magnetic field. Supplementary Figure 12b shows that the Josephson tunneling evolves to be a dome-like shape under a small magnetic field of 0.4 mT. Strikingly, for sample S3 with initially AB-type $I_c(T)$ at zero field (Fig. 2c2), such a dome-like behavior can be induced by a small magnetic field (Supplementary Fig. 12c). Similar behaviors were reported in intrinsic Bi-2212 Josephson junctions[57]. There, the increase in $I_c(T)$ with increasing temperature was attributed to the depinning and realignment of the flux pancakes. We speculate that trapped fluxes can play a similar role in junctions even at zero magnetic field. They can produce the non-monotonic behavior that resembles the predicted feature of a topologically non-trivial origin.

Finally, we speculate on the mechanism for the gradual temperature variation of $B_0$ in the Fraunhofer patterns (Fig. 3c, further summarized in Supplementary Fig. 12d). It indicates a crossover from the tunneling between two layered superconductors to that between anisotropic ones. The crossover occurs at a temperature $T^*$ that $\xi_c(T) = \xi_{c0}/\sqrt{1 - T/T_c}$ becomes comparable to the interlayer distance between the neighboring $CuO_2$ planes. At $T < T^*$, $\xi_c$ is shorter than the interlayer distance such that the twisted junction should be treated as a multilayered system. Such a layered system can be described by the phenomenological theory proposed by Lawrence and Doniach[58]. If $w < \lambda_j$ ($\lambda_j$ is the Josephson penetration depth) and there exist an infinite number of layers with equal Josephson currents, the L-D model gives rise to the Fraunhofer pattern with a period[59]: $B_{L-D} = \Phi_0/(wd)$. Here $d$ is the junction thickness and in Bi-2201 it should be equal to $d_{Cu-Cu}$. At $T^* < T < T_c$, $\xi_c$ is large enough that the top and bottom Bi-2201 can be individually treated as an anisotropic superconductor following the 3D Ginzburg-Landau (G-L) formula[7]. The Fraunhofer period becomes:

$B_{G-L} = \Phi_0 / \left[ w \left( d_{\mathrm{Sr-Sr}} + 2\lambda_c \right) \right]$, where $\lambda_c = \lambda_0 / \sqrt{1 - T/T_c}$ is the $c$-axis penetration depth and $d_{\mathrm{Sr-Sr}}$ is the thickness of the tunnel barrier (between two SrO planes that sandwich the double BiO planes). This theoretical analysis can qualitatively reproduce the experimental trend (inset of Supplementary Fig. 12d).

In summary, we successfully realize twisted cuprate bilayers out of a previously unexplored compound—Bi-2201. We demonstrate that our samples host ultraclean interfaces and preserve the stoichiometry from the bulk—critical for investigating their intrinsic properties. Transport experiments reveal pronounced $I_c R_n$ values in these high-quality junctions at the twist angle of 45°, similar to that observed in twisted Bi-2212 junctions. These results show sharp contrast to the pure $d$-wave scenario and strongly suggest the presence of an isotropic pairing component in the twisted junctions. We carry out a systematic investigation to check the validity of recent theoretical proposals on the emergent $d + id$ wave. We show that a Josephson junction at $45.0 \pm 0.2°$ possesses: (1) conventional temperature dependence of the critical current; (2) standard Fraunhofer pattern with no frequency doubling as a function of temperature; (3) integer Fiske steps. These results strongly disfavor the existence of Cooper pair co-tunneling. Our results shed valuable insight into the twisted cuprate systems and help settle down the debate on their underlying pairing symmetry.

## Methods

The optimally doped $Bi_2Sr_{1.63}La_{0.37}CuO_{6+x}$ (Bi-2201) single crystal was grown by the traveling solvent floating zone method[60]. The crystal possesses an onset superconducting transition temperature ($T_c$) of 32 K. All junctions included in this work were fabricated from the same piece of crystal (Supplementary Fig. 10). The polydimethylsiloxane (PDMS) used for the van der Waals stacking was prepared by using Dow Corning Sylgard 184, mixing base and curing agent in a ratio of 10:1. It was then transferred onto a clean sapphire slide and quickly baked to 120 °C on a hot plate for 10 min for adhesion. The sapphire slide ensured good thermal conduction at cryogenic temperatures. The $SiO_2$/Si wafers (4 inch) with a 285 nm thick $SiO_2$ were used as substrates. For prepatterning the substrates, the entire wafer was processed by employing the ultraviolet photo-lithography followed by metal deposition of Ti/Au (5/40 nm) with E-beam. Both the pristine and prepatterned wafers were then diced into rectangular pieces ($3 \times 10$ mm$^2$) for fabrication of the Josephson junctions (schematically shown in Fig. 1b). Before loading into the glovebox for junction stacking, the unpatterened/patterned substrates were treated by $O_2$ plasma (power 100 W/60 W, pressure 0.1/0.3 mbar, time period 1/3 min), in order to enhance the bonding between Bi-2201 and $SiO_2$[5]. The sample fabrication was carried out in a glovebox filled with Ar gas ($H_2O < 0.01$ ppm, $O_2 < 0.01$ ppm).

For samples S1 and S2, we employed FIB with gallium ions to pattern the narrow strips. The acceleration voltage was set at 30 kV and a small beam current of 2 pA was used to minimize the degradation effect. It was reported in the Bi-2212 intrinsic junctions that there existed a 30 nm thick amorphous layer after the FIB processing[61]. This thickness is one order of magnitude smaller than the width of our samples (3.5 μm for S1 and 5 μm for S2) such that the influence of the amorphous layer should be negligible. However, in other strips with their widths smaller than 2 μm, we observed that $T_c$ of the junction became suppressed to 10 K–15 K. It indicates that when the strip is too narrow FIB patterning may cause oxygen out-diffusion, presumably due to heating. We emphasize that this effect is minimized for samples S1 and S2, because their $T_c$ values are the same as that of the bulk crystal. Nevertheless, we observe no hysteresis in the $I$-$V$ characteristics of S1 and weaker hysteresis in that of S2, in contrast to those of samples without FIB processing. To understand this difference, we start from the RCSJ model[62]. Whether the hysteresis is pronounced or not depends on the damping factor: $\beta_c = 2e I_c R_n^2 C / \hbar$, where $C$ is the

effective capacitance of the junction. A large $\beta_c \gg 1$ corresponds to pronounced hysteresis. The equation can be further written as:

$$\beta_c = 2e(I_c R_n) \rho_n \epsilon_r \epsilon_0 l / d / \hbar, \qquad (1)$$

where $\rho_n$ is the normal state resistivity, $l$ is the effective length of the junction, $d$ is the effective distance between the two superconducting planes that constitute a planar capacitor (essentially $l \sim d$), and $\epsilon_r$ is the relative dielectric constant. From the TEM images (Fig. 1), the atomic structure of the junctions is unchanged after FIB patterning such that $l$ and $d$ should not be affected. A less pronounced hysteresis, i.e., a smaller $\beta_c$, may therefore be attributed to the reduction of $I_c R_n$, $\rho_n$ or $\epsilon_r$. It indicates that FIB patterning may influence our junctions in a subtle manner[63]. Future improvements can be gained by utilizing helium ions for FIB patterning[64,65].

The samples were wired to the chip carriers within an hour at ambient condition outside the glovebox. Transport measurements were carried out in two different closed-cycle helium-free systems equipped with superconducting magnets (base temperature below 1.6 K). Resistances were measured by the standard low-frequency lock-in technique with a typical ac current of 1 μA (13.3 Hz). For $I$-$V$ measurements, we used a dc source to supply the current and a nanovoltmeter to measure the voltage in a four-terminal configuration. We measured $dV/dI$ simultaneously by measuring the four-terminal ac voltage with a lock-in amplifier at an ac current of 500 nA (13.3 Hz). To measure the Fraunhofer pattern, the samples were mounted on a home-built piezo-driven rotator inside the cryogenic system. The rotator has an angular precision of 0.006°. To align the magnetic field to the in-plane direction of the junction, we rotated the sample at 1 T and 14 K until the junction resistance showed a minimum. We ramped up the magnetic field from zero in one direction and swept the tunneling current at each fixed magnetic field to measure both $V$ and $dV/dI$. To obtain the data in the opposite magnetic field direction, we carried out a thermal cycling up to 50 K and back at zero field in order to expel trapped fluxes.

After the transport measurements, the junctions were investigated by transmission electron microscope (TEM) (S1–S5). The TEM samples were prepared by using FIB technique with Ga ions. For S1 and S2, HAADF-STEM experiments were carried out in the FEI-Titan Cubed Themis 60–300 operating at 300 keV with double spherical aberration (Cs) correctors and a spatial resolution of 0.6 Å. For S3–S5, TEM experiments were performed in the JEOL-ARM200CF operating at 200 keV with double spherical aberration (Cs) correctors and a spatial resolution of 0.8 Å. To determine the twist angles accurately, we focused the electron beam on either top or bottom Bi-2201 to observe their respective Kikuchi patterns. We then calculated the angle based on the separation between the two patterns.

## Data availability

The data generated in Figs. 1–5 of this study have been deposited in: https://doi.org/10.6084/m9.figshare.22760870. All other data that support the plots within this paper are available from the corresponding author upon reasonable request.

## Code availability

The computer code used for data analysis is available upon request from the corresponding author.

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

## Acknowledgements

We thank fruitful discussions with Federico Paolucci, Richard Klemm, and Philip Kim on the manuscript. This work is financially supported by the National Natural Science Foundation of China [Grants No. 52388201 (Q.-K.X.), No. 12274249 (D.Z.), No. 12141402 (Y.Z.), No. 12004041 (Y.Z.), No. 11888101 (X.J.Z.), No. 52072400 (Q.Z.), No. 51991344 (L.G.), and No. 52025025 (L.G.)]; Ministry of Science and Technology of China [2021YFA1401800 (X.J.Z.)]; Innovation Program for Quantum Science and Technology [Grant No. 2021ZD0302600 (Y.Z.)]; Beijing Natural Science Foundation [Grant No. Z190010 (L.G.)].

## Author contributions

Y.Z., D.Z., and Q.-K.X. conceived the project. H.W. and Y.Z. fabricated the devices and carried out the transport measurements. Z.B. carried out the FIB patterning. Z.W., X.L., Q.Z., L.G., and J.Z. carried out TEM experiments. X.H. and S.H. helped in constructing the cryogenic stage. H.-Y.X. did the theoretical calculations. J. Cu., M.H., J. Ch. assisted in transport measurements. Y.D., L.Z., and X.J.Z. grew the single crystals. H.W., Y.Z., and D.Z. analyzed the data. D.Z. and Y.Z. wrote the paper with the input from Q.-K.X. All authors discussed the results and commented on the manuscript.

## Competing interests

The authors declare no competing interests.
