## [Peer Review File · Nature Communications]

Prominent Josephson tunneling between twisted single copper oxide planes of $\text{Bi}_2\text{Sr}_{2-x}\text{La}_x\text{CuO}_{6+y}$REVIEWER COMMENTS

Reviewer #1 (Remarks to the Author):

The manuscript of Heng Wang et al. reports transport measurements in twisted thin cuprate Bi-2201 with atomically clean tunneling interfaces. Their results suggested the pairing mechanism is isotropic rather than the long-thought d-wave type. Their fabrication and structural characterization support that their devices are of state-of-the-art quality. They gave three points against unconventional d-wave pairing mechanism: 1) In a few clean samples, traditional temperature-dependent critical current behaviors are observed, which disfavors the theoretical predictions of novel superconductivity on the interface. They trivialized other samples that show untraditional temperature dependence by comparing them to the ones in a finite magnetic field. 2) Only integer Fiske steps are observed. 3) The observed prominent Josephson tunneling cannot follow a $\cos(2\theta)$ trend because it will give rise to an unphysical gap and critical current for the intrinsic junction.

This work is along the line of recent efforts on using advanced van der Waals fabrication to explore the possibility of novel topological superconductivity but, surprisingly, as a testbed to question the pairing mechanism of cuprates. To the best of my knowledge, this is the first work about twisted thin Bi-2201 with single CuO₂ planes and will attract interest from a broad audience, such as 2D materials, nanodevice physics, and superconductivity. This work may warrant an expedited publication in Nature Communications with minor changes. However, the author seems to overinterpret some of the results, as detailed below:

1. The authors use Fiske steps (self-resonance effect) rather than Shapiro steps (rf irradiation), commonly used for the phase-sensitive ac-JJ experiment. In Shapiro steps, the controllability of frequency and quantization of the step can unambiguously assign steps number. However, in Fiske steps, one can arbitrarily assign an integer or half integer-integer step, e.g., in fig.4c, it could be 1/2, 1(a), 3/2(b). Can the author figure out from the device geometry what the resonance frequency should be? A discussion, e.g., a comparison between Fiske steps vs. Shapiro steps, can help the broad audience to understand the features better.
2. Figure 5 a,b only uses two devices (S1 and S2) to estimate the gap and critical current, which could be misleading. Though it's a zero-parameter fitting, the uncertainty of the twist angle can give rise to overestimations. Figure 5c could deviate from the s-wave isotropic model.
3. Figure 6 d shows very few data points showing the saturation behavior of L-D models. Such data quality cannot support the speculation to become a claim.

These speculations (Figs.5-6) are well-suited in text format in the discussion section. But any of this interpretation in figure format may only suit supplements.

Please also consider the following suggestions and questions to improve the manuscript:

- 1) It's worth mentioning in the main text that the STEM is done after FIB and transport measurement.
- 2) What are the supermodulations observed in STEM? Can the author elaborate more and provide any references? If lattice relaxation is involved as a function of twist angle, this may be a piece of evidence.
- 3) Why are the data in Fig2 c1-c4 significantly noisier? Did the author try the measurement without a surrounding superconducting magnet to avoid flux trapping in the magnet? Related to this, why in Fig3, when $T=18K$ and $22K$, does the data with $B || \sim < 0.1T$ become so clean?
- 4) In figure 4c, can the author try to differentiate or amplify the data $B || > 0.4$ to reveal the $k = 1$ mode more clearly?
- 5) Can the author estimate the contact resistance of Au-BSCCO from the data of S3 or intrinsic sample? My worry is that contact heating may contribute.

Reviewer #2 (Remarks to the Author):

See an attached pdf file

Reviewer #3 (Remarks to the Author):

In this article, the authors report the realization of tunneling Josephson junctions between two flakes of the $\text{Bi}_2\text{Sr}_{2-x}\text{La}_x\text{CuO}_{6+y}$ ($\text{Bi}2201$) high- T_c superconducting cuprate, twisted by an angle of around 45° . It has been known for almost 30 years from the experiments by Tsuei and Kirtley, and Wollman et al., that in cuprates, the superconducting order parameter has mostly a d-wave symmetry (dx^2-y^2) with nodes at 45° between the k_x and k_y axis. However, the existence of an additional component of small amplitude, has also been debated for many years without reaching a true conclusion. The interest for this question was renewed recently with the theoretical prediction that a chiral topological state could form in 45° twisted $\text{Bi}2212$ cuprate flakes that can be thought of as an emergent $dx^2-y^2 \pm idxy$ superconductor. In the present paper, the authors observe a significant Josephson current in 45° twisted junctions, and conclude that it results from the presence of an additional isotropic s-wave pairing component. Indeed, in the case of a pure dx^2-y^2 order parameter, the Josephson current should be strictly zero. Overall, the manuscript presents interesting results with a large set of experiment data regarding structural characterizations and transport measurements. In particular, I appreciate the experimental approach based on an impressive degree of control of the Van der Waals stacking method

even I also notice that several articles (including by the authors themselves) addressing the same issue have already been published on twisted Van der Waals cuprates superconductors. Beyond some weakness in the data analysis which are listed below, my main concern is related to the conclusion of the study. Although I acknowledge the intention to bring an answer to the important questions of the pairing symmetry in the cuprates, I must say that, unfortunately, I am somewhat disappointed with the conclusion as it seems to me that we didn't learn much more on this old debate. The results may be relevant in the specific context of twisted Van der Waals Josephson junction but the authors do not propose any model to support their claim. Finally, the results are rather similar to that reported in twisted junctions, made with Bi-2212, a cuprate of the very same family. Under these circumstances, I cannot give my recommendation to publish this article in Nat. Commun. A revised version may be considered for a more specialized journal. I list below a series of comments and questions that the authors may want to consider in a future submission.

1) The determination of the angle using the Kikuchi patterns as in reference 6 should be presented in the manuscript. I would also appreciate to see a close up on the atomic structure in Figure 1. Directions of observation should also be specified for each sample.

2) The authors mentioned a presence of a surmodulation in some of their samples but do not explain its origin. What is the reason for that? Given the importance of the surmodulation amplitude (even if it seems to be reduced at the top layer), it is very unlikely that superconductivity is not affected at all. In particular, I expect that the symmetry of the order parameter could be significantly modified. This may explain why the Josephson current remains significant at a 45° twist. More generally, the authors should take into account the effect of disorder in the prediction of the angle dependence of the junction properties.

3) I found the following sentence misleading: "More importantly, the two rotated CuO₂ planes exhibit the same intensities as those CuO₂ planes far away from the interface, attesting to the uncompromised quality of the two superconducting layers involved in the Josephson tunneling process". The fact that CuO₂ planes display the same intensities in the integrated HAADF-STEM image is not sufficient to guarantee that the crystal quality is uncompromised.

4) As each junction has a different area, I would recommend to plot the current density (and critical current density J_c) instead of the absolute current value.

5) As mentioned by the authors, the current-voltage characteristics seem to exhibit different types of behavior depending on the sample. Some of them show an abrupt switching with a more or less pronounced hysteresis while others show a smooth RCSJ-like characteristic. The authors should comment on such variety of behaviors. On general grounds, switching and hysteresis result either from capacitive effect or from thermal runaway. Is the switching stochastic?

6) Information on the geometry should be given for the Fraunhofer experiment. What is the corresponding area defining the magnetic flux in the junction. As presented, the figure is quite unclear and the agreement remains only qualitative with the model. What is the origin of the additional structures that are seen in the color plots below 10K ?. Instead of plotting dI/dV in color code, I would also recommend to plot directly $I_c(B)$ for a better comparison with the model.

7) Figure 5 summarizes the claims of the authors. In fig 5a, the authors extrapolate the value of $I_c R_n$ and J_c at 0 angle from the values measured on samples S1 and S2 based on the expected angle dependence for the pure d-wave case. As they found that the extrapolated values are too large (compared to the gap energy for instance) they ruled out the pure d-wave model. I do not think that such extrapolation from 2 data points measured at the 45° is justified. A tiny error in the determination of these data points would result in a huge error in the extrapolated values.

8) In Figure 5b, the value of the $I_c R_n$ for a 45° twist reaches 1 mV which is a fraction of the value expected for a 0° twist. This value is too high to be ascribed to an intrinsic s-wave pairing component in the Bi2201 cuprates. For instance ARPES experiments show that if there is an additional pairing component in cuprates it is very tiny.

Assuming that the Josephson coupling measured in this experiment is not related to some kind of disorder, it may result from the emergence of a pairing component which is specific to this type of artificial Van der Waals junction. This is interesting but more work is needed to support this claim, in particular the authors must propose an explanation for the emergence of the isotropic pairing component.

Reviewer #1 (Remarks to the Author):

The manuscript of Heng Wang et al. reports transport measurements in twisted thin cuprate Bi-2201 with atomically clean tunneling interfaces. Their results suggested the pairing mechanism is isotropic rather than the long-thought d-wave type. Their fabrication and structural characterization support that their devices are of state-of-the-art quality. They gave three points against unconventional d-wave pairing mechanism: 1) In a few clean samples, traditional temperature-dependent critical current behaviors are observed, which disfavors the theoretical predictions of novel superconductivity on the interface. They trivialized other samples that show untraditional temperature dependence by comparing them to the ones in a finite magnetic field. 2) Only integer Fiske steps are observed. 3) The observed prominent Josephson tunneling cannot follow a $\cos(2\theta)$ trend because it will give rise to an unphysical gap and critical current for the intrinsic junction.

This work is along the line of recent efforts on using advanced van der Waals fabrication to explore the possibility of novel topological superconductivity but, surprisingly, as a testbed to question the pairing mechanism of cuprates. To the best of my knowledge, this is the first work about twisted thin Bi-2201 with single CuO₂ planes and will attract interest from a broad audience, such as 2D materials, nanodevice physics, and superconductivity. This work may warrant an expedited publication in Nature Communications with minor changes.

[Our reply] We thank the reviewer for the very positive remark of our work. We have carried out new experiments and made substantial revisions to address the critical comments of the reviewer. We provide a point-to-point answer below.

However, the author seems to overinterpret some of the results, as detailed below:

1. The authors use Fiske steps (self-resonance effect) rather than Shapiro steps (rf irradiation), commonly used for the phase-sensitive ac-JJ experiment. In Shapiro steps, the controllability of frequency and quantization of the step can unambiguously assign steps number. However, in Fiske steps, one can arbitrarily assign an integer or half integer-integer step, e.g., in fig.4c, it could be 1/2, 1(a), 3/2(b). Can the author figure out from the device geometry what the resonance frequency should be? A discussion, e.g., a comparison between Fiske steps vs. Shapiro steps, can help the broad audience to understand the features better.

[Our reply] Following the reviewer's nice suggestion, we have provided the following description in the revised manuscript about the similarity and differences of Fiske steps and Shapiro steps: "Typically, the AC Josephson effect is demonstrated as current steps at discrete voltages—Shapiro steps—in a junction under microwave irradiation. Here, no external microwave source is necessary because we utilize the rectangular shape of our junction. Such a cavity possesses certain resonance modes which can be excited under a bias voltage. Interference between the electromagnetic fields resonating in the cavity and the AC Josephson effect of the junction gives rise to another type of current steps at discrete voltages—Fiske steps".

The reviewer is correct that the Fiske steps alone cannot distinguish between tunneling with Cooper pairs ($2e$) and tunneling that purely arises from paired Cooper pairs with $4e$. However, a

purely $4e$ tunneling has a specific temperature dependence of the Josephson critical current, which we exclude by showing the standard $I_c(T)$ in Fig. 2c1-c3. We have added the following texts in the paragraph on the Fiske steps to clarify the issue:

“We point out that the monotonic increase of the sequence of Fiske steps alone is compatible with Josephson tunneling of either purely $2e$ or purely $4e$ carriers. However, a dominant higher-order tunneling with $4e$ has a specific temperature dependence, which we have excluded by the conventional $I_c(T)$ in Fig. 2. The remaining possibility is that there exists a mixture of a dominant $2e$ -tunneling with a small fraction of $4e$ -tunneling. In this case, the Fiske steps have two sets: one prominent set at integer steps due to $2e$ -tunneling and one weaker set at half integers due to $4e$ -tunneling. The intensity in dI/dV of the Fiske steps would therefore show non-monotonic evolution with consecutively increasing bias voltages, in sharp contrast to our data in Fig. 4.”

2. Figure 5 a,b only uses two devices (S1 and S2) to estimate the gap and critical current, which could be misleading. Though it's a zero-parameter fitting, the uncertainty of the twist angle can give rise to overestimations. Figure 5c could deviate from the s-wave isotropic model.

[Our reply] We have taken the reviewer's suggestion (in Point 3) and removed Fig. 5a,b. The calculations in those figures are not strictly fitting but a way to show the disagreement between experiment and the pure d -wave pairing in another perspective. We have revised the corresponding texts to clarify this point.

As for Figure 5c (Fig. 5 in the revised manuscript), the reviewer is correct that the angular dependence suggested by our experimental data seems to deviate from the model for incoherent tunneling of isotropic s -wave pairing. However, a completely angular independent critical current only arises from incoherent tunneling of s -wave pairing, in which the in-plane momentum of the Cooper pairs is not preserved. Once the tunneling becomes coherent, the orbital effect may become important. It can give rise to angular dependent critical current even for purely s -wave pairing. A recent Josephson tunneling experiment on twisted NbSe_2 —a conventional s -wave superconductor—showed just this angular dependence. We discussed this possibility in the Discussion section and cited the NbSe_2 paper as ref. 53. Besides, the angular dependence in our experiment may indicate a mixture of s -wave and d -wave pairing. We therefore conclude in the Discussion section that our work excludes the scenario of purely d -wave pairing in the twisted junctions.

3. Figure 6 d shows very few data points showing the saturation behavior of L-D models. Such data quality cannot support the speculation to become a claim.

These speculations (Figs.5-6) are well-suited in text format in the discussion section. But any of this interpretation in figure format may only suit supplements.

[Our reply] We have moved Fig. 6 to the supplementary information. Figure 5 has also been simplified.

Please also consider the following suggestions and questions to improve the manuscript:

1) It's worth mentioning in the main text that the STEM is done after FIB and transport

measurement.

[Our reply] We thank the reviewer for this and the following constructive suggestions, which greatly help improve the clarity of the paper.

We have added a sentence in the main text to mention that the STEM is done after transport measurements.

2) What are the supermodulations observed in STEM? Can the author elaborate more and provide any references? If lattice relaxation is involved as a function of twist angle, this may be a piece of evidence.

[Our reply] We have revised the corresponding texts and added explanations on the supermodulations: “For samples S1, S3 and S4, wavy undulations can be observed in the bottom halves of the images, suggesting that these sections are viewed from the $[0\bar{1}0]$ crystalline direction perpendicular to the uniaxial supermodulations running along the $[100]$ direction. The top parts show no wavy structures because they are viewed from the $[1\bar{1}0]$ direction that is along the diagonal between $[0\bar{1}0]$ and $[100]$. The supermodulations are usually attributed to the misfit between the CuO_2/SrO planes and the BiO plane [46,47] and can locally modulate the energy gap [48].” Here, refs. [46-48] are the newly added references that study the supermodulation in experiment and theory, respectively.

Figure R1 Thickness of 0.5 UC of Bi-2201 in the top and bottom flakes. Data points are extracted from the peak positions of the TEM intensity profile along the c -axis. Abscissae show the layer number. Black arrows indicate 0.5 UC just above or below the twisted interface. Error bars are from the pixel resolution in the TEM image.

As for the possible change in the lattice structure as a function of twist angle, we have carried out further analysis of the STEM images. We show in Fig. R1 that the thickness of half-unit-cell above or below the twist boundary stays the same as that deep inside the flake. We have also compared the intensity profile of the CuO_2 planes at the twist boundary and in the bulk in Fig. S3. Both analyses indicate that the crystalline structure at the interface stays the same as that in the bulk. Since our flakes are relatively thick—80-200 nm, it is unlikely that the twisted interface can result in changes in the lattice structure of the individual flakes.

3) Why are the data in Fig2 c1-c4 significantly noisier? Did the author try the measurement without a surrounding superconducting magnet to avoid flux trapping in the magnet? Related to this, why in Fig3, when $T=18\text{K}$ and 22K , does the data with $B_{\parallel} \sim < 0.1\text{T}$ become so clean?

[Oure reply] So far, all the experiments were carried out in cryogenic systems equipped with superconducting magnets. We speculate that the noise in Fig. 2c1-c4 arises from stochastic fluctuation of the critical current. We have carried out further experiments to elucidate this point. We repeatedly measure the critical current (1000 repetitions) on one junction at fixed temperatures. Figure R2 shows the histogram. With decreasing temperature, the distribution seems to broaden first and then shrinks. This non-monotonic behavior was previously seen in intrinsic Josephson junctions [*Phys. Rev. Lett.* **99**, 037002 (2007)] and is likely related to the crossover from thermally activated regime to the macroscopic quantum coherent regime. A detailed study requires measurements with over 10000 repetitions for a set of temperatures, which is beyond the scope of the present paper.

Figure R2 Histograms of critical Josephson current in a 30° twisted sample. Here we measure the 1000 I - V characteristics at each temperature. This junction has a superconducting transition temperature of 30 K.

For the noises seen in Fig. 3, we originally provided an explanation in the figure caption. We have now added the following explanation in the main text: “In addition, it is notable that noises in the color plot become suddenly prominent upon increasing B_{\parallel} to a certain value (marked by the red arrows). This onset field becomes larger with decreasing temperatures, reminiscent of the

temperature dependence of critical field (Fig. S10). We therefore attribute the noises to the entrance of Abrikosov vortices above the lower critical field. They are created by a small out-of-plane magnetic field due to slight misalignment between the magnetic field and the sample plane.”

4) In figure 4c, can the author try to differentiate or amplify the data $B \parallel > 0.4$ to reveal the $k = 1$ mode more clearly?

[Our reply] We thank the reviewer for the kind suggestion. We have revised the figure by amplifying the data at 0.55 and 0.554 T.

5) Can the author estimate the contact resistance of Au-BSCCO from the data of S3 or intrinsic sample? My worry is that contact heating may contribute.

[Our reply] We have taken further experiment in the configuration illustrated in Fig. R3 and estimated the contact resistance to be around 200 Ohm. We believe this contact resistance together with the relatively small current (typically < 1 mA) we apply would not cause significant heating. As a reference, we measured in the past FeSe/SrTiO₃ samples with a typical resistance of 100 Ohm [see, for example, *Appl. Phys. Lett.* **108**, 202603 (2016)]. The heating only becomes noticeable as the applied current exceeds 10 mA at temperatures below 10 K.

To further clarify this issue, we have fabricated another junction and studied the I - V characteristics with different sweeping rates. We observe that the variation in the critical current measured with different rates is comparable to the statistical fluctuation of the critical current at a fixed rate. Therefore, the heating effect seems to be negligible.

Figure R3 Schematic of the circuit employed for measuring the contact resistance. The resistance of wires from the room-temperature side to the sample is subtracted.

Reviewer #2 (Remarks to the Author):

This manuscript reports the presence of an isotropic pairing component in the twisted Bi2201 junctions with a twist angle close to 45 degrees, showing a sharp contrast to the conventional d-wave scenario in the cuprate superconductors. The authors very carefully prepared the twisted junctions with atomically flat interfaces and employed an on-site cold stacking method to fabricate the junctions. If the obtained results are attributed to the intrinsic properties of Josephson junctions, this study will pose important reconsideration to the understanding of the cuprates.

[Our reply] We thank the reviewer for appreciating the importance of our work.

However, all of the experimental results seem to show that the fabricated junctions never form Josephson junctions, as described in the following comments. In my understanding, the obtained I-V characteristics clearly indicate that the critical current is not determined by Josephson current but depinning or depairing current in the junction (or an upper flake). I agree that the authors succeeded in preparing the twisted junctions with ultraclean interfaces and uncompromised stoichiometry. Unfortunately, this study shows that such smoothly connected bilayer junctions are not always appropriate for the demonstration of Josephson effects. In conclusion, I do not recommend this manuscript for the publication in Nature Communications. The authors should revise the manuscript, addressing the following comments.

[Our reply] We do not agree with the reviewer that our data can be explained by the depinning or de-pairing effect. There exist crucial differences between the two effects and the Josephson effect. First of all, only a Josephson junction hosts Fraunhofer pattern under a magnetic field that modulates the phase difference across the junction. In comparison, the depinning or de-pairing current depends monotonically on the magnetic field due to the weakening of the superconducting state. Secondly, the AC Josephson effect is the defining property of Josephson junction. This is usually demonstrated by irradiating the junction with microwave such that Shapiro steps occur in the I-V characteristic. Alternatively, the AC Josephson effect manifests itself as Fiske steps in a junction that is shaped as a cavity. Under a bias voltage, the cavity can be resonantly excited and the electromagnetic wave associated with the cavity induces AC Josephson effect too, even in the absence of externally applied microwave.

Figure 3 provides unambiguous evidence for both Fraunhofer pattern and Fiske steps, fully underpinning the Josephson nature of our junctions. To better appreciate these features, we plot in Fig. R4 below the extracted critical current and the higher order branch (Fiske steps). Contrary to the reviewer's assignment of the higher order branch to hysteresis (Point 2), Fig. R5 shows that there exists no hysteresis in the I-V characteristic. It is crystal clear in this plot that the additional branch stems from a current step—Fiske step—at a finite voltage. We stress that the data plotted in Fig. 4, in which the Fiske steps are acknowledged by the reviewer (demonstrated from the reviewer's statement in Point 4), are the same data set in Fig. 3 but plotted differently.

After establishing that what we observed is the Josephson effect, we further argue that this Josephson effect stems from the twisted interface. We highlight that the twisted interface is made of the overlapping section between the top and bottom flakes. This overlapping area is much smaller than the size of each individual flake. This contrasting tunneling areas help distinguish the tunneling at the twisted interface from that in the bottom flake. A clear comparison between these two critical currents is given in Fig. S7. The intrinsic Josephson tunneling within the top flake does not contribute since our contacts are placed on the bottom.

Figure R4 Fraunhofer pattern and Fiske steps of sample S1. **a.** Red curves represent critical Josephson currents at zero bias from the I-V characteristics at different $B_{||}$. Black curves trace out the local maximum of dV/dI in the colored plot of Fig. 3. Purple shades indicate the regions for Fiske steps. **b.** I-V characteristic at selected $B_{||}$. The chosen magnetic fields are marked in panel **a** by the vertical bars. Vertical lines mark the voltage positions for the Fiske steps identified in the colored plot of Fig. 4.

Figure R5 I-V characteristics of sample S1 at a selected in-plane magnetic field. This particular magnetic field corresponds to the situation where the main branch in the color-coded plot bifurcates. It clearly demonstrates that the branch at a higher current (about $15 \mu A$) in Fig. 3c is at non-zero bias. We use circles (curve) to show the data in the positive (negative) sweeping directions. The overlap suggests that there exists no hysteresis.

1. In Figs.2b2 to 2b4, the hysteresis loop in the I-V characteristics is not limited in the underdamped Josephson junctions with a finite capacitance. It is quite strange that the magnitude of the hysteresis for a positive current is always smaller than that for a negative current. This suggests a suspicion that the observed hysteresis is

attributed to the Joule heating in the resistive state, which often occurs in a small superconducting bridge. In addition, the magnitude of critical current is also asymmetrical about a current inversion. Such a non-reciprocal behavior can be induced by the difference of a surface barrier for a vortex intrusion between top and bottom surfaces in the upper flake. A similar effect is recently reported by Mizuno et al. (IEEE, Trans. Appl. Supercond. vol.32, 6601005 2022). The scattered behavior in the temperature dependence of I_c at low temperatures (See Figs. 2c2-2c4) is also due to the non-reciprocal response. In this study, the I-V characteristics were obtained by using a dc source. The Joule heating effects can be checked by changing a current-sweep rate or using a pulse method. In my experience, the plots of V/I versus I at low T become very similar to those of R versus T with a small I , in the case of Joule heating.

[Our reply] We note that it is purely coincidental that the positive current shown in Fig. 2b2 to 2b4 is smaller. For the same sample, sweeping current several times gives rise to fluctuating critical currents with either a larger or smaller positive current, which is demonstrated in Fig. R6 and Fig. R7 for samples S3 and S4.

Figure R6 I-V characteristics of sample S3 obtained at 1.6 K (a) and 10 K (b). Here the data at 10 K is obtained from a different run than that shown in Fig. 2b2. The critical current in the positive direction is larger in this run.

Figure R7 I-V characteristics of sample S4 at 1.6 K (a) and 8 K (b) and sample OP1 at 1.6 K (c).

Figure R8 *I-V characteristics of a 45°-twisted junction at 1.6 K (a,b) and 5 K (b) with the specified sweeping rates. The superconducting transition of this junction is 26 K. The first row of panels show that the critical Josephson current has slight variation at the same temperature and sweeping rate due to statistical fluctuations. The second row of panels indicate that decreasing the sweeping rate by a factor of 8 does not introduce noticeable change in the critical current.*

Following the reviewer's comment, we have carried out new experiments by changing the current-sweep rate. The collected data are summarized in Fig. R8. We observe that the variation in the critical current measured with different rates is comparable to the statistical fluctuation of the critical current at a fixed rate. Therefore, the heating effect is negligible.

2. In Fig. 3 (sample S1), the color mappings of dV/dI show two branches in a range between $B=0.05$ T and 0.1 T below 10 K, corresponding to the appearance of the hysteresis in the $I-V$ characteristics. In this case, an upper branch is closer to the true critical current, rather than a lower branch. Thus, the experimental results do not agree with theoretical lines. Rather, the critical current is weakly dependent on B up to ~ 0.1 T, and is suddenly decreased above 0.15 T. Figure S2 (sample S2) of Supplementary information also shows a similar behavior, showing a disagreement with the theoretical Fraunhofer pattern above 0.1 T. These results can be easily explained when the critical current is determined by the depinning current for a vortex induced by the self-field due to the superconducting current flow.

[Our reply] As shown in Fig. R5, there is no hysteresis in the $I-V$ characteristics in the specified regimes for sample S1. The additional branches pointed out by the reviewer occur at non-zero voltages such that the corresponding current is not the Josephson critical current. We realize that the color plot may provide a false impression for the magnetic field dependence of the critical current. To avoid this confusion, we have added a panel (Fig. 3b) to show the extracted critical Josephson current as a function of the in-plane magnetic field.

The branches that the reviewer pointed out are related to the emergence of Fiske steps. We have added additional discussion in the texts to emphasize this point.

3. According to the estimation by the authors, the lateral sizes of the twisted junction seem to

be always larger than Josephson penetration depth. In this case, the phase difference of the superconducting wave function between the upper and lower flakes is not homogeneous in the junction plane, leading to asymmetrical Fraunhofer patterns. However, the results for sample S1 and S2 show symmetrical behavior about magnetic field inversion.

[Our reply] The calculated Josephson penetration depth only provides an estimation that can guide the design of the real junction. Whether the phase difference is homogeneous or not should be eventually determined by the experimental result rather than the theoretical calculation. Therefore, the observation of symmetric Fraunhofer patterns provides clear evidence for a homogeneous phase difference. We note that previous experiments on intrinsic Josephson tunneling also demonstrated standard Fraunhofer patterns in samples with lateral sizes larger than the Josephson penetration depth (Fig. R9). The Fraunhofer pattern gradually becomes less discernible with an increasing lateral size of the junction (Fig. R9b).

Y. I. Latyshev, et al, *Physic C* 362, 156 (2001) Y. I. Latyshev, et al, *Phys. Rev. Lett.* 77, 932 (1996)

Figure R9 Fraunhofer patterns obtained from previous experiments on the intrinsic Josephson junctions of Bi-2212. Figures are clipped from the listed references. Red numbers indicate the Josephson penetration depth and the physical size of the junction(s).

4. In Fig. 4, the magnetic field where Fiske steps were observed is similar to that for the intrinsic Josephson junctions (IJJs) reported in Ref. [47], rather than for the conventional Josephson junctions where Fiske steps were observed at much smaller magnetic field. This is related to a thickness of the Josephson junction, indicating that the thickness of the twisted junction is almost the same as that of IJJs. This also suggests that the observed Fiske steps cannot be distinguished with those appearing in the IJJs of the upper flake, because the electrode configuration in this study is almost the same as that to measure the response of the upper flake. Thus, Fig. 4 does neither prove the formation of Josephson junction or show inconsistency with an interpretation that the critical current is determined by the depinning or depairing current in the upper flake.

[Our reply] We agree with the reviewer that the junction thickness in our twisted samples is similar to IJJ. However, the tunneling areas are markedly different. Previous experiments on IJJ were done on FIB patterned samples with small sizes comparable to the overlapping area of our samples. By contrast, in our experiment, the tunneling area for IJJ is much larger than that of the twisted interface.

5. In Fig. 5, the unphysically large values of I_{cRn} and J_c extrapolated to 0 degree strongly support

that the fabricated junctions do not show Josephson effects. Such large values are derived from the results for the twisted junctions with ~ 45 degrees where the $I_c R_n$ product cannot be explained by the superconducting gap, in contrast to Ambegaokar-Baratoff (AB) theory for the tunnel junction.

[Our reply] We hope that our above-mentioned replies with new experimental data have convinced the reviewer that our junctions show Josephson coupling at the twisted interface. Starting from this, the unphysically large values from the extrapolation constitutes strong evidence that our experimental results cannot be explained by the standard $\cos(2\theta)$ dependence for purely d -wave pairing. This is one of the core messages we point out in the Discussion section.

6. As for the temperature dependence of I_c near T_c , there are two types in the experimental results. One shows a positive curvature near T_c showing a larger increase than AB theory (Fig. 2c3), while the other shows a negative curvature near T_c (Fig. 2c4, Fig.6 and Figs. S4g, S4m). The former behavior is characteristic for the depairing current, proportional to $(1-T/T_c)^{1.5}$ near T_c , and the latter is similar to the behavior of the depinning current. The unusual decrease of I_c at low temperature shown in Fig. 6 seems to be explained by vortex creep.

[Our reply] As we stated in the manuscript, the AB formula is derived for the tunneling between two conventional s -wave superconductors with the same superconducting gaps. In reality, the cuprate superconductor does not fit into the conventional category. Moreover, cuprates host intrinsic inhomogeneity of the superconducting gaps. It may result in gap asymmetry between the two areas for each tunneling event. In general, the deviation from the rather simple model of AB theory cannot be taken as evidence for the absence of Josephson tunneling.

7. Perhaps, toward improved experiments, the existence of an insulating surface due to stoichiometric disorder may be necessary for an ideal formation of Josephson junction. However, it is expected that the precise control of the junction property is quite difficult. Another important point is the downsizing of the lateral area. FIB etching is a good tool to obtain a small area of the junction.

[Our reply] We thank the reviewer for the kind suggestions. We have included two more references on the recent progresses in successful patterning of cuprate superconductors into nanowires by using helium ion FIB (ref. [62] and [63]).

Reviewer #3 (Remarks to the Author):

In this article, the authors report the realization of tunneling Josephson junctions between two flakes of the $\text{Bi}_2\text{Sr}_{2-x}\text{La}_x\text{CuO}_{6+y}$ ($\text{Bi}2201$) high- T_c superconducting cuprate, twisted by an angle of around 45° . It has been known for almost 30 years from the experiments by Tsuei and Kirtley, and Wollman et al., that in cuprates, the superconducting order parameter has mostly a d-wave symmetry (dx^2-y^2) with nodes at 45° between the k_x and k_y axis. However, the existence of an additional component of small amplitude, has also been debated for many years without reaching a true conclusion. The interest for this question was renewed recently with the theoretical prediction that a chiral topological state could form in 45° twisted $\text{Bi}2212$ cuprate flakes that can be thought of as an emergent $dx^2-y^2 \pm dxy$ superconductor. In the present paper, the authors observe a significant Josephson current in 45° twisted junctions, and conclude that it results from the presence of an additional isotropic s-wave pairing component. Indeed, in the case of a pure dx^2-dy^2 order parameter, the Josephson current should be strictly zero. Overall, the manuscript presents interesting results with a large set of experiment data regarding structural characterizations and transport measurements. In particular, I appreciate the experimental approach based on an impressive degree of control of the Van der Waals stacking method even I also notice that several articles (including by the authors themselves) addressing the same issue have already been published on twisted Van der Waals cuprates superconductors. Beyond some weakness in the data analysis which are listed below, my main concern is related to the conclusion of the study. Although I acknowledge the intention to bring an answer to the important questions of the pairing symmetry in the cuprates, I must say that, unfortunately, I am somewhat disappointed with the conclusion as it seems to me that we didn't learn much more on this old debate. The results may be relevant in the specific context of twisted Van der Waals Josephson junction but the authors do not propose any model to support they claim. Finally, the results are rather similar to that reported in twisted junctions, made with $\text{Bi}2212$, a cuprate of the very same family. Under these circumstances, I cannot give my recommendation to publish this article in *Nat. Commun.* A revised version may be considered for a more specialized journal.

[Our reply] We thank the reviewer for appreciating our work. The reviewer is correct that the conclusion from our present work is unchanged in comparison to that of our previous report. However, given the significance of this debate, a deeper understanding can only become possible with more materials and more precisely controlled experiments. This is achieved by our present work by closing the loopholes of previous works: (1) our junctions have substantially improved crystalline quality in comparison to all previous reports. We present a comparison of the atomically resolved images in Fig. R10. Notably, all previous works show decreased signal intensity at the twisted interface and substantially expanded interlayer distance with sometimes a trilayer of BiO as the barrier. In our previous work, we discussed that such a triple BiO layer allows antisymmetric term in the tunneling matrix element, which may complicate the data interpretation. Such an uncertainty is completely removed in our present work. (2) we are able to demonstrate a slew of high-quality data all in single samples ($S1$ and $S2$): high crystalline quality with atomic resolution, a twist angle very close to 45 degrees, Josephson tunneling at different temperatures and as a function of in-plane magnetic field, leading to Fraunhofer pattern and Fiske steps. Such a demanding combination was not achieved before and it removes any doubt that a large Josephson

tunneling does occur in 45°-twisted junctions. (3) the twisted Bi-2201 junctions has not been studied before.

Figure R10 TEM images of the twisted cuprate junctions in the published works and this work.

We also point out that the present work addresses not only the “old” debate over *s*-wave vs. *d*-wave but also the rising new debate on the emergent topological superconductivity in twisted cuprates [see doubts raised by theorists in: *Phys. Rev. B* **105**, L201102 (2022), *Phys. Rev. B* **105**, 245127 (2022)]. Our work is timely because the proposal of *d+id*-wave pairing has attracted enormous interest, leading to further prediction of high temperature Majorana zero modes [*Phys. Rev. Lett.* **128**, 137002 (2022), highlighted by *Physics*]. Our present work seriously challenges this proposal by providing four important pieces of experimental evidence against it. Three of them are collected in junctions with twist angles close to 45 degrees (44.8° and 44.0° in samples S1 and S2): 1) conventional temperature dependences of the critical current; 2) standard Fraunhofer patterns without doubling in frequency as a function of temperature; 3) Fiske steps with integer numbers in the sequence (in S1). A fourth piece of evidence against it is the observation of non-monotonic temperature dependence of the critical current even at 0 degree. All these observations are against the theoretical predictions for the *d+id*-wave pairing.

We have studied more samples and carried out further analysis to address the points raised by the reviewer. We elaborate in the following points.

I list below a series of comments and questions that the authors may want to consider in a future submission.

1) The determination of the angle using the Kikuchi patterns as in reference 6 should be

presented in the manuscript. I would also appreciate to see a close up on the atomic structure in Figure 1. Directions of observation should also be specified for each sample.

[Oure reply] We thank the reviewer for this very constructive remark. We realize that the quality of atomically resolved images degraded due to figure compression. The atomic resolution can be appreciated in the uncompressed figure, which we submit this time. For a better appreciation of the atomic structure, we have prepared a separate supplementary file that contains the uncompressed raw TEM images. We have also added a new section in the supplementary information (supplementary note 2) to describe how we use the Kikuchi patterns to determine the twist angle.

2) The authors mentioned a presence of a surmodulation in some of their samples but do not explain its origin. What is the reason for that ? Given the importance of the surmodulation amplitude (even if it seems to be reduced at the top layer), it is very unlikely that superconductivity is not affected at all. In particular, I expect that the symmetry of the order parameter could be significantly modified. This may explain why the Josephson current remains significant at a 45° twist. More generally, the authors should take into account the effect of disorder in the prediction of the angle dependence of the junction properties.

[Our reply] All of our samples possess the same supermodulation since they are all cleaved from the same single crystal. We have revised the corresponding texts on the supermodulations in the manuscript: “For samples S1, S3 and S4, wavy undulations can be observed in the bottom halves of the images, suggesting that these sections are viewed from the $[0\bar{1}0]$ crystalline direction perpendicular to the uniaxial supermodulations running along the $[100]$ direction. The top parts show no wavy structures because they are viewed from the $[1\bar{1}0]$ direction that is along the diagonal between $[0\bar{1}0]$ and $[100]$. The supermodulations are usually attributed to the misfit between the CuO_2/SrO planes and the BiO plane [46][47] and can locally modulate the energy gap [48]”. Here, ref. [46-48] are the newly added references that study the supermodulation in experiment and theory, respectively.

We emphasize that the supermodulations are not emergent ordering of our twisted system. They are intrinsic in Bi-2201 and Bi-2212 single crystals. The reviewer is correct that such a uniaxial modulation can affect superconductivity on the local level, as revealed in the newly added ref. [48]. We also agree with the reviewer that the supermodulations break the C_4 rotational symmetry and should influence the pairing symmetry as well. Until now, no experiments reveal such a symmetry breaking effect on the order parameter. We have added this speculation in the Discussion section.

For the disorder effect, we note that the shaded bands in Fig. 5 take into account the fluctuating values of $I_c R_n$ at zero or 90 degrees twist angles. They account for the variation from sample to sample. We argue that such a disorder effect should also follow the $\cos(2\theta)$ dependence of d -wave pairing because the symmetry argument is valid on the local level. Namely, the tunneling at different sites may be different due to disorder, but they all get suppressed as a function of twist angle, demanded by the d -wave pairing.

3) I found the following sentence misleading : “ More importantly, the two rotated CuO_2 planes exhibit the same intensities as those CuO_2 planes far away from the interface, attesting to the uncompromised quality of the two superconducting layers involved in the Josephson tunneling process”. The fact that CuO_2 plans display the same intensities in the integrated HAADF-STEM image is not sufficient to guarantee that the crystal quality is uncompromised.

[Our reply] The reviewer is correct that the integrated intensity does not sufficiently guarantee high crystalline quality. As explained in the reply to Point 1, the TEM images in the previous version of the manuscript were blurred due to compression. In the original TEM images now submitted as a separate file in the revised manuscript, we show that the signal intensities from individual atoms are comparable. A more quantitative comparison is made in Fig. R11 below (Fig. S3). Here, we compare the intensity profiles taken at the CuO_2 planes at the interface and in the bulk. They show nearly overlapping peaks, demonstrating the high crystalline quality at the interface.

Figure R11 Intensity analysis of the CuO_2 plane. a & c, High-resolution TEM images of S1 and S2. b & d, Intensity profiles of the CuO_2 plane in the upper flake close to the interface (indicated by the red arrows in a and b) and that in the bulk (indicated by the black arrows in a and b).

4) As each junction has a different area, I would recommend to plot the current density (and critical current density J_c) instead of the absolute current value.

[Our reply] We thank the reviewer for this suggestion. We have added the tunneling area for each

sample to make readers aware of the varying sizes. We note that the actual tunneling area may be smaller than the apparent overlapping area.

5) As mentioned by the authors, the current-voltage characteristics seem to exhibit different types of behavior depending on the sample. Some of them show an abrupt switching with a more or less pronounced hysteresis while others show a smooth RCSJ-like characteristic. The authors should comment on such variety of behaviors. On general grounds, switching and hysteresis result either from capacitive effect or from thermal runaway. Is the switching stochastic ?

[Our reply] In fact, we commented on the difference in the manuscript:” For samples S3, S4 and OP1, the I - V characteristics show jumps from the zero-bias branch to the branch with finite resistances. There also exists prominent hysteresis in the two opposite sweeping directions. For samples S1 and S2 (Fig. S2b), however, the switching seems continuous. These two samples are patterned by focused ion beam (FIB) into long strips (insets of Fig. 2a and Fig. S5a) for the investigation of Fraunhofer patterns (to be discussed in the following section). This additional step seems to suppress the hysteresis.”

We have provided the following analysis in the Materials and methods section:

“...we observe no hysteresis in the I - V characteristics of S1 and weaker hysteresis in that of S2, in contrast to those of samples without FIB processing. To understand this difference, we start from the RCSJ model [60]. Whether the hysteresis is pronounced or not depends on the damping factor: $\beta_c = 2eI_c R_n^2 C / \hbar$, where C is the effective capacitance of the junction. A large $\beta_c \gg 1$ corresponds to pronounced hysteresis. The equation can be further written as:

$$\beta_c = 2e(I_c R_n) \rho_n \epsilon_r \epsilon_0 l / d / \hbar,$$

where ρ_n is the normal state resistivity, l is the effective length of the junction, d is the effective distance between the two superconducting planes that constitute a planar capacitor (essentially $l \sim d$), and ϵ_r is the relative dielectric constant. From the TEM images (Fig. 1), the atomic structure of the junctions is unchanged after FIB patterning such that l and d should not be affected. A less pronounced hysteresis, i.e., a smaller β_c , may therefore be attributed to the reduction of $I_c R_n$, ρ_n or ϵ_r . It indicates that FIB patterning may influence our junctions in a subtle manner [61].”

We have carried out further experiments to demonstrate the stochastic nature of the switching. The histogram of the critical current shows thermal broadening, consistent with that observed in intrinsic Josephson junctions (Fig. R12). We have also investigated if the hysteresis is caused by thermal heating by sweeping the tunneling current with different rates (Fig. R13). We observe that the variation in the critical current measured with different rates is comparable to the statistical fluctuation of the critical current at a fixed rate. Therefore, the heating effect is negligible.

Figure R12 Histograms of critical Josephson current in a 30° twisted sample. Here we measure the 1000 I-V characteristics at each temperature. This junction has a superconducting transition temperature of 30 K. With decreasing temperature, the distribution seems to broaden first and then shrinks. This non-monotonic behavior was previously seen in intrinsic Josephson junctions [Phys. Rev. Lett. **99**, 037002 (2007)] and is likely related to the crossover from thermally activated regime to the macroscopic quantum coherent regime. A detailed study requires measurements with over 10000 repetitions at each fixed temperature, which is beyond the scope of the present paper.

Figure R13 I-V characteristics of a 45° -twisted junction at 1.6 K (a,b) and 5 K (b) with the specified sweeping rates. The superconducting transition of this junction is 26 K. The first row of panels show that the critical Josephson current has slight variation at the same temperature and sweeping rate due to statistical fluctuations. The second row of panels indicate that decreasing the sweeping rate by a factor of 8 does not introduce noticeable change in the critical current.

6) Information on the geometry should be given for the Fraunhofer experiment. What is the corresponding area defining the magnetic flux in the junction. As presented, the figure is quite unclear and the agreement remains only qualitative with the model. What is the origin of the

additional structures that are seen in the color plots below 10K ?. Instead of plotting dI/dV in color code, I would also recommend to plot directly $I_c(B)$ for a better comparison with the model.

[Our reply] Following the reviewer's suggestion, we have added $I_c(B)$ plots together with an illustration of the measurement configuration in the revised Fig. 3. The additional structures that the reviewer pointed out are related to Fiske steps that we discuss in Fig. 4. We have added more texts to make this connection clearer. For convenience, we show the correspondence between the features in the color plot and the I - V characteristic below.

Figure R14 Fraunhofer pattern and Fiske steps of sample S1. **a.** Red curves represent critical Josephson currents at zero bias from the I - V characteristics at different B_{\parallel} . Black curves trace out the local maximum of dV/dI in the colored plot of Fig. 3. Purple shades indicate the regions for Fiske steps. **b.** I - V characteristic at selected B_{\parallel} . The chosen magnetic fields are marked in panel **a** by the vertical bars. Vertical lines mark the voltage positions for the Fiske steps identified in the colored plot of Fig. 4.

7) Figure 5 summarizes the claims of the authors. In fig 5a, the authors extrapolate the value of I_{cRn} and J_c at 0 angle from the values measured on samples S1 and S2 based on the expected angle dependence for the pure d -wave case. As they found that the extrapolated values are too large (compared to the gap energy for instance) they ruled out the pure d -wave model. I do not think that such extrapolation from 2 data points measured at the 45° is justified. A tiny error in the determination of these data points would result in a huge error in the extrapolated values.

[Our reply] We have removed Fig. 5a,b in the revised manuscript. The purpose of this way of analysis is to highlight the disagreement between experiment and pure d -wave pairing from another perspective. We have revised the corresponding texts in the Discussion section to clarify this issue. Essentially, the data points at around 45 degrees stand as counter-examples to the pure d -wave scenario. Forcing a $\cos(2\theta)$ dependence upon them would yield absurd values at zero degrees. Conversely, one can take the experimental values measured at 0 degree and calculate the expected values at 44 or 44.8 degrees following the $\cos(2\theta)$ dependence. The resulted values (as suggested by the blue band in Fig. 5) can be substantially smaller than the experimentally measured values at 44 and 44.8 degrees.

8) In Figure 5b, the value of the $I_c R_n$ for a 45° twist reaches 1 mV which is a fraction of the value expected for a 0° twist. This value is too high to be ascribed to an intrinsic s-wave pairing component in the Bi2201 cuprates. For instance ARPES experiments show that if there is an additional pairing component in cuprates it is very tiny. Assuming that the Josephson coupling measured in this experiment is not related to some kind of disorder, it may result from the emergence of a pairing component which is specific to this type of artificial Van der Waals junction. This is interesting but more work is needed to support this claim, in particular the authors must propose an explanation for the emergence of the isotropic pairing component.

[Our reply] Indeed, the large $I_c R_n$ indicates the presence of a strong s-wave pairing component. Our experiment is able to directly probe the phase of the superconducting wavefunction: the absence of Josephson coupling at 45° twist stems from the sign-change property of the d-wave pairing. By contrast, ARPES, as far as we understand, only probes the amplitude of the wavefunction. Therefore, ARPES seems ineffective in distinguishing between d-wave pairing and an anisotropic s-wave pairing. Furthermore, both ARPES and STM probes the CuO_2 layer through the charge reservoir layers of BiO and SrO. We would like to bring the reviewer's attention to some works by probing directly on the CuO_2 layer of cuprates [*Phys. Rev. Lett.* **89**, 087002(2002); *Science Bulletin* **61**, 1239 (2016); *National Science Review* **9**, nwab225(2022)] without the capping of BiO/SrO layers. There, the tunneling spectra show a clear U-shaped gap, in sharp contrast to the V-shaped gap that is usually observed when probing through the BiO/SrO surface layers. We therefore suspect that the presence of BiO/SrO layers in the sample studied by ARPES may hinder the identification of an isotropic pairing component.

As stated in the reply to Point 2, we have added our speculation on the prominent isotropic pairing component. We note that the present work focuses on establishing the existence of such an isotropic pairing component on a firm ground and excluding the proposal of higher-order tunneling. This is achieved by a demanding combination of experiments on the same samples. Our work will definitely instigate future theoretical developments.

REVIEWER COMMENTS

Reviewer #1 (Remarks to the Author):

This is my second read of the manuscript about the twisted thin Bi-2201 tunneling junctions. Their experiment clearly shows the atomically clean interface of the junction, strongly supports what they saw is from this synthetic interface rather than an intrinsic junction, and disapproves of the pure d-wave pairing picture. There is no clear signature of topological superconductivity. Although, in some sense, this work is a type of `negative` report, this work will set a strong constraint for theoretical work on twisted high T_c and guides future experiments on searching for topological superconductivity and the continuing debate on the pairings of cuprates.

In the revised version, the authors adequately addressed my concerns and removed the arbitrariness of their interpretation. As I can tell, they have already reached a milestone regarding the sample quality. The experiment is impressive. Their data is abundant to support their claim. I strongly recommend its publication in Nature Communications.

Reviewer #2 (Remarks to the Author):

See an attached pdf file

Reviewer #3 (Remarks to the Author):

The authors have significantly revised the manuscript and provided detailed answers to the reviewers' comments. While I am not fully convinced by the conclusions of the work, I recognize that the experimental results are quite impressive and provide new insights on the issue of pairing symmetry in cuprates. For this reason, I recommend the publication of the article in Nature Communications.

I agree that the authors responded my comments to the previous manuscript as honestly as possible. However, my conclusion was not changed. I still suspect that the observed critical current is attributed to Josephson effect, as described below. Thus, I do not recommend this manuscript for the publication in Nature Communications.

1. As for the reply for the 1st comment of my report, the authors showed the related results in Fig. R6 to R8. I agree that the observed hysteresis could not be explained by a simple hypothesis that they were due to Joule heating occurring in the resistive state. However, the hysteretic behavior in the I-V characteristics is also observed in phase slip phenomena, as reported by S. Michotte et al. (See Phys. Rev. B 69, 094512, 2004). Note that phase slips also show Josephson-like phenomena such as Shapiro steps for microwave irradiation and the interference effects under magnetic field. Although the authors stated that the observed variation in the critical current was attributed to the stochastic switching to the resistive state, the switching to the voltage state due to phase slips is also stochastic, as studied by D. Pekker et al. (See Phys. Rev. B 80, 214525, 2009). The author showed the switching current distribution at three temperatures in Fig. R2 or R12, as the reply to the comments from other reviewers. I found that the peak current in the histograms measured at each temperature showed non-monotonous temperature dependence, in contrast to the conventional behaviors (simply increasing with decreasing temperature) in the switches in Josephson junctions and phase slips in the superconducting wire (or films). This strongly indicates that there are at least more than two origins in the switches to the voltage state. That is, the critical current at $T=1.6$ K is determined by another origin than that at $T=10$ K. Thus, the authors should discuss the origin of the critical current in the measured data more carefully.
2. The authors stated that the observation of Fiske steps became an important proof of Josephson junction together with data shown in Fig. R4. In Fig. R4 (b), however, I found several small voltage steps above the critical current at $B=0$ T, suggesting the occurrence of phase slip lines (or phase slip centers). In this case, the observed voltage steps at finite magnetic field can be interpreted as phase slips under magnetic field. Thus, the color mappings of dV/dI in Fig. 2(c) are very misleading. They should be replaced by the critical current measured at zero bias. The authors should discuss whether a periodic modulation of the critical current as a function of in-plane magnetic field is explained by phase slips or Josephson effects. In addition, in the 4th reply, the authors referred to a markedly difference of junction area between the overlapping part of the device and each part of two flakes. If the voltage steps under the in-plane magnetic field are not Fiske steps, such a difference of junction area is unimportant.

Reviewer #1 (Remarks to the Author):

This is my second read of the manuscript about the twisted thin Bi-2201 tunneling junctions. Their experiment clearly shows the atomically clean interface of the junction, strongly supports what they saw is from this synthetic interface rather than an intrinsic junction, and disapproves of the pure d-wave pairing picture. There is no clear signature of topological superconductivity. Although, in some sense, this work is a type of `negative` report, this work will set a strong constraint for theoretical work on twisted high Tc and guides future experiments on searching for topological superconductivity and the continuing debate on the pairings of cuprates. In the revised version, the authors adequately addressed my concerns and removed the arbitrariness of their interpretation. As I can tell, they have already reached a milestone regarding the sample quality. The experiment is impressive. Their data is abundant to support their claim. I strongly recommend its publication in Nature Communications.

[Our reply] We would thank the reviewer for the concise and accurate summary of our work. We would also like to extend our gratitude to the reviewer for the strong recommendation for publication in Nature Communications.

Reviewer #2 (Remarks to the Author):

I agree that the authors responded my comments to the previous manuscript as honestly as possible. However, my conclusion was not changed. I still suspect that the observed critical current is attributed to Josephson effect, as described below. Thus, I do not recommend this manuscript for the publication in Nature Communications.

[Our reply] The reviewer proposed two other mechanisms than the Josephson effect for generating hysteretic I - V characteristics: (a) phase slip centers (PSCs) in superconducting nanowires (as discussed in the two references mentioned by the reviewer in Point 1); (b) phase slip lines (PSLs) in superconducting thin films (as mentioned by the reviewer in Point 2). Here we draw the three situations, including the Josephson effect, in Fig. R1 below. For the Josephson junction, we focus on the planar configuration: two superconducting planes separated by a layer of insulator/normal metal. This is the geometry for c -axis cuprate Josephson junctions because the CuO_2 layers host superconductivity and they are separated by non-superconductive BiO/SrO layers.

A superconducting nanowire requires that the sample width is comparable to the superconducting coherence length (Fig. R1a). In cuprates, the coherence length is on the order of 1 nm. By contrast, our samples have a typical width of 10 μm . The twisted Bi-2201 junctions are clearly not superconducting nanowires, thus they do not host PSCs.

For a superconducting strip with a width much larger than the superconducting coherence length, PSLs may form and give rise to hysteretic I - V characteristic. The PSL is formed by fast running vortices—so-called kinematic vortices—with their motion transverse to the current direction [A. Andronov *et al.*, *Physica C* **213**, 193(1993); G. R. Berdiyrov *et al.*, *Phys. Rev. B* **79**, 174506 (2009)]. This motion is induced by the Magnus force experienced by each Abrikosov vortex under a current flow, as indicated in Fig. R1b. After the formation of a PSL, the superconductor is separated into two regions, effectively forming an in-plane Josephson junction. The Josephson nature was indeed demonstrated through the observation of Shapiro steps by irradiating microwave [A. G. Sivakov, *et al.*, *Phys. Rev. Lett.* **91**, 267001 (2003)]. However, applying an in-plane magnetic field along the current direction cannot induce any phase difference between the two superconducting regions. An in-plane magnetic field therefore cannot result in Fraunhofer interference pattern in the case

Fig. R1 Schematic illustrations of the phase slip center in a superconducting wire, the phase slip line in a superconducting film, and the planar Josephson junction

of a superconducting thin film hosting a PSL. By contrast, we applied an in-plane magnetic field parallel to the two-dimensional superconducting planes of CuO_2 layers and observed clear Fraunhofer interference pattern. The observation of Fraunhofer pattern in our junctions patently distinguishes the Josephson tunneling out-of-plane from the PSL induced effect in the plane.

The reviewer stated that “phase slips also show ... the interference effects under magnetic field”. To the best of our knowledge, we are not aware of any experimental observation of Fraunhofer patterns in a superconductor with either PSC or PSL. A. Jindal *et al.* studied PSL effect in exfoliated flakes of Bi-2212. They observed only a monotonic decrease of the critical current [Sci. Rep. 7, 3295 (2017)]. The absence of interference pattern under a magnetic field is also supported by simulation. G. R. Berdiyrov *et al.* demonstrated that the critical current for a superconducting strip with PSL decreases monotonically with the increasing magnetic field [Phys. Rev. B 79, 174506 (2009)]. By “interference effects under magnetic field”, the reviewer might be referring to the oscillations seen in the paper: A. G. Sivakov *et al.*, Phys. Rev. Lett. 91, 267001 (2003). However, there, a SQUID was intentionally constructed by etching a window in the center of a superconducting strip. The situation is distinctly different from a uniformly connected superconductor and the critical current only oscillates in a small range without reaching zero. For convenience, we show a clip from their work below.

FIG. 1. Current-voltage (I - V) characteristics of a $15\ \mu\text{m}$ wide Sn strip with a $5\ \mu\text{m} \times 15\ \mu\text{m}$ window in the middle shown in inset (a). Insets (b) and (c) present laser scanning microscope (LSM) images taken at points 1 and 2 of the I - V curve, respectively. White arrows indicate the phase-slip lines (PSLs).

FIG. 4. Critical current I_c versus magnetic field H for $25\ \mu\text{m}$ wide Sn strips with different window sizes: (a) $(5 \times 5)\ \mu\text{m}^2$; (b) $(5 \times 10)\ \mu\text{m}^2$; (c) $(5 \times 15)\ \mu\text{m}^2$.

Fig. R2 Clips from A. G. Sivakov *et al.*, Phys. Rev. Lett. 91, 267001 (2003)

Another clear distinction between the PSL effect and the Josephson effect is the number of jumps in the I - V characteristics. For a superconductor that is tens of micrometer long, multiple PSLs appear as the current increases. The emergence of each PSL gives rise to a jump in the I - V characteristic. We show two recent examples in Fig. R3 below. The data are from exfoliated flakes with lateral sizes that are comparable to those of our twisted cuprate junctions. By contrast, we have measured several twisted cuprate junctions in a wide current range (Fig. R4). Only a single

Fig. R3 Clips from the references that show typical I - V characteristics induced by PSL. The work on the left studied exfoliated flakes of NbSe_2 . The work on the right studied exfoliated flakes of Bi-2212 .

Fig. R4 I - V characteristics of two twisted Bi-2201 junctions, showing a single jump in the positive or negative current direction in a wide current range.

jump appears in the positive/negative direction—a clear manifestation of Josephson tunneling just between the two twisted CuO_2 planes. We have added the discussion on these two different types of I - V characteristics in the main text. We have also added Fig. R4 to the supplementary information.

In summary, the proposal of PSL fails to account for the different aspects of our data in a consistent manner. In addition, we would like to point out that the current is forced to flow vertically in our artificial junction. By contrast, the PSL induced effect happens in-plane. If the hysteretic behaviors seen in all our junctions were from the PSL induced effect, it would suggest that no Josephson behavior occurs in the vertical transport. That is to say, the coupling from a CuO_2 plane across the BiO/SrO layers to the other CuO_2 plane is not Josephson coupling, in contrast to the understanding from decades of studies on c -axis transport of Bi-2212 and Bi-2201 (refs. [38-44]).

1. As for the reply for the 1st comment of my report, the authors showed the related results in Fig. R6 to R8. I agree that the observed hysteresis could not be explained by a simple hypothesis

that they were due to Joule heating occurring in the resistive state. However, the hysteretic behavior in the I-V characteristics is also observed in phase slip phenomena, as reported by S. Michotte et al. (See Phys. Rev. B 69, 094512, 2004). Note that phase slips also show Josephson-like phenomena such as Shapiro steps for microwave irradiation and the interference effects under magnetic field. Although the authors stated that the observed variation in the critical current was attributed to the stochastic switching to the resistive state, the switching to the voltage state due to phase slips is also stochastic, as studied by D. Pekker et al. (See Phys. Rev. B 80, 214525, 2009). The author showed the switching current distribution at three temperatures in Fig. R2 or R12, as the reply to the comments from other reviewers. I found that the peak current in the histograms measured at each temperature showed non-monotonous temperature dependence, in contrast to the conventional behaviors (simply increasing with decreasing temperature) in the switches in Josephson junctions and phase slips in the superconducting wire (or films). This strongly indicates that there are at least more than two origins in the switches to the voltage state. That is, the critical current at $T=1.6$ K is determined by another origin than that at $T=10$ K. Thus, the authors should discuss the origin of the critical current in the measured data more carefully.

[Our reply] We believe we have adequately distinguished our Josephson junctions from superconductors with phase slip centers/lines in the previous point.

Here, we mainly address the concern about the non-monotonic temperature dependence. The reviewer is correct that the peak current in the histogram is: 1.81 mA at 10 K; 1.85 mA at 5 K; 1.84 mA at 1.6 K. However, the decrease is as small as 0.01 mA from 5 K to 1.6 K and falls within the statistical spread of the critical current (Full width at half-maximum: 0.03 mA). As we stated in the reply to the other reviewers, the histogram is constructed from 1000 measurements and is mainly to demonstrate the statistical fluctuation. Whether there indeed exists a drop in the critical current with decreasing temperature or not requires statistics with measurements over 10000 repetitions, as was done in ref. [39]. In fact, the statistics we did before only took into account the switching behavior in the positive current direction. In Fig. R5 below, for the same data set, we take into

Fig. R5 Histograms of critical Josephson current in a 30° twisted sample. Here we measure the I-V characteristic for 1000 repetitions. Switching events in both positive and negative current directions are taken into account.

account switching events in both positive and negative directions. It effectively doubles the total number of events that we consider in the histogram. Clearly, the peak currents at 1.6 K and 5 K get closer and fall within the statistical spread.

2. The authors stated that the observation of Fiske steps became an important proof of Josephson junction together with data shown in Fig. R4. In Fig. R4 (b), however, I found several small voltage steps above the critical current at $B=0$ T, suggesting the occurrence of phase slip lines (or phase slip centers). In this case, the observed voltage steps at finite magnetic field can be interpreted as phase slips under magnetic field. Thus, the color mappings of dV/dI in Fig. 2(c) are very misleading. They should be replaced by the critical current measured at zero bias. The authors should discuss whether a periodic modulation of the critical current as a function of in-plane magnetic field is explained by phase slips or Josephson effects. In addition, in the 4th reply, the authors referred to a markedly difference of junction area between the overlapping part of the device and each part of two flakes. If the voltage steps under the in-plane magnetic field are not Fiske steps, such a difference of junction area is unimportant.

[Our reply] The “steps” at $B=0$ T are caused by the electrical noise in the measurement. We show a zoom-in plot in Fig. R6. The increasing slope gets disrupted by the fluctuation of only one or two data points. This is distinctly different from the situation of steps caused by phase slip lines, as represented in Fig. R3.

It is also impossible that the Fiske steps we observed in Fig. 4 is related to those fluctuating signal at $B = 0$ T. We show in Fig. R7b that the fluctuating signal, that the reviewer found as voltage steps by phase slippage, occurs randomly upon sweeping the current multiple times. This is in sharp contrast to the stable voltage steps we observed under in-plane magnetic field.

Fig. R6 I - V characteristics of sample S1 at different temperatures (panel a) or in repeated measurements at a fixed temperature (panel b). Data are horizontally offset in panel a and vertically offset in panel b for clarity.

Reviewer #3 (Remarks to the Author):

The authors have significantly revised the manuscript and provided detailed answers to the reviewers' comments. While I am not fully convinced by the conclusions of the work, I recognize that the experimental results are quite impressive and provide new insights on the issue of pairing symmetry in cuprates. For this reason, I recommend the publication of the article in Nature Communications.

[Our reply] We thank the reviewer for recommending our paper for publication in Nature Communications.

REVIEWERS' COMMENTS

Reviewer #2 (Remarks to the Author):

The authors very clearly responded to my comments in the 2nd report. I agree with the authors' explanation about the periodic modulation of the critical current as a function of in-plane magnetic field, which cannot be explained by the phase slip scenario. Thus, the results shown in Fig. 3 in the manuscript are certainly attributed to the superconducting current along the c axis. In addition, the results shown in Fig. R4 of the 3rd response also rule out the possibility of an occurrence of PSL in the fabricated devices. Although the electrical noise in the measurements seems to extraordinarily influence the results of I-V characteristics and the distribution of the switching current to the voltage state, I conclude that the present manuscript and the authors' responses to the reviewers' comments satisfactorily meet the criteria of publication in Nature Communications. While the uncertainty is remained in the measured results, I agree that this work provide renewed interest to an issue of paring symmetry in twisted cuprate junctions. I recommend the publication in Nature Communications.

Reviewer #2 (Remarks to the Author):

The authors very clearly responded to my comments in the 2nd report. I agree with the authors' explanation about the periodic modulation of the critical current as a function of in-plane magnetic field, which cannot be explained by the phase slip scenario. Thus, the results shown in Fig. 3 in the manuscript are certainly attributed to the superconducting current along the c axis. In addition, the results shown in Fig. R4 of the 3rd response also rule out the possibility of an occurrence of PSL in the fabricated devices. Although the electrical noise in the measurements seems to extraordinarily influence the results of I-V characteristics and the distribution of the switching current to the voltage state, I conclude that the present manuscript and the authors' responses to the reviewers' comments satisfactorily meet the criteria of publication in Nature Communications. While the uncertainty is remained in the measured results, I agree that this work provide renewed interest to an issue of paring symmetry in twisted cuprate junctions. I recommend the publication in Nature Communications.

[Our reply] We thank the reviewer for recommending our paper for publication.